# One Model to Translate Them All: Universal Any-to-Any Translation for Heterogeneous Collaborative Perception

**Yang Li** [* 1]  **Weize Li** [* 1]  **Quan Yuan** [1]  **Congzhang Shao** [1]  **Guiyang Luo** [1]  **Yunqi Ba** [1]  **Xuanhan Zhu** [1]  **Xinyuan Ding** [1]  **Xiaoyuan Fu** [1]  **Jinglin Li** [1]

## Abstract

By sharing intermediate features, collaborative perception extends each agent's sensing beyond standalone limits, but real-world feature modality heterogeneity remains a key barrier to effective fusion. Most existing methods, including direct adaption and protocol-based transformation, typically rely on training adapters for newly emerging feature modalities and often require additional retraining or fine-tuning. Such repeated training is costly and is often infeasible across manufacturers due to model and data privacy constraints, limiting real-world scalability. To address this issue, we propose `UniTrans`, a universal any-to-any feature modality translation model that instantiates translators on the fly for arbitrary modalities. `UniTrans` pretrains a bank of translator expert parameters and learns their combination coefficients as a function of source-to-target modality mapping. The mapping is measured in a modality-intrinsic latent space, where an intrinsic encoder extracts modality-specific yet scene-invariant codes from single-frame intermediate features, enabling `UniTrans` to instantiate translators in a zero-shot manner. Experiments on OPV2V-H and DAIR-V2X demonstrate that `UniTrans` consistently outperforms state-of-the-art methods in both simulated and real-world settings, enabling efficient any-to-any translation through a universal model. The code is available at https://github.com/CheeryLeeyy/UniTrans.

*Equal contribution [1]State Key Laboratory of Networking and Switching Technology, Beijing University of Posts and Telecommunications, Beijing, China. Correspondence to: Quan Yuan <yuanquan@bupt.edu.cn>.

*Proceedings of the 43rd International Conference on Machine Learning*, Seoul, South Korea. PMLR 306, 2026. Copyright 2026 by the author(s).

## 1. Introduction

Intermediate fusion-based collaborative perception has become a cornerstone of next-generation autonomous driving (Chen et al., 2024), as sharing bird's-eye-view (BEV) features mitigates single-vehicle limitations such as occlusions and limited long-range perception range (Wang et al., 2020). Yet, in practical deployments, diverse sensor configurations and perception architectures yield intermediate features in different modalities (Bai et al., 2024). Newly introduced agents with such heterogeneous features can induce pronounced cross-agent domain shifts, posing a major barrier to effective collaboration and often degrading performance (Lu et al., 2024).

Existing studies on heterogeneous collaborative perception aim to bridge these gaps and can be broadly grouped into two paradigms: one-to-one adaptation and two-step adaptation, as illustrated in Fig. 1(a) and (b). One-to-one adaptation methods (e.g., MPDA (Xu et al., 2023), PnPDA (Luo et al., 2025)) perform domain transfer by training a customized adapter for each *modality mapping, i.e., a specific transformation from the source feature space to the target feature space*. While effective, this paradigm often requires repeated adapter retraining for new or evolving modality mappings, incurring substantial cost. Two-step adaptation alleviates this burden by introducing a unified protocol space. With the protocol as an intermediate bridge, a new agent only needs to learn a mapping to the protocol space, avoiding many pairwise adapters. However, protocol-based methods such as STAMP (Gao et al., 2025) and NegoCollab (Shao et al., 2025) rely on a pre-defined or negotiated protocol space that may not be well-suited to newly emerging modality types. In practice, accommodating new modalities often requires protocol adjustment, which necessitates relearning the induced modality mappings and incurs additional training overhead.

Overall, both paradigms inevitably rely on repeated rounds of joint training or fine-tuning to preserve collaboration performance, and the integration cost quickly escalates as more emerging agents with heterogeneous modalities are introduced (Hu et al., 2025; Luo et al., 2025). Moreover, frequent cross-manufacturer joint training is often impractical due to

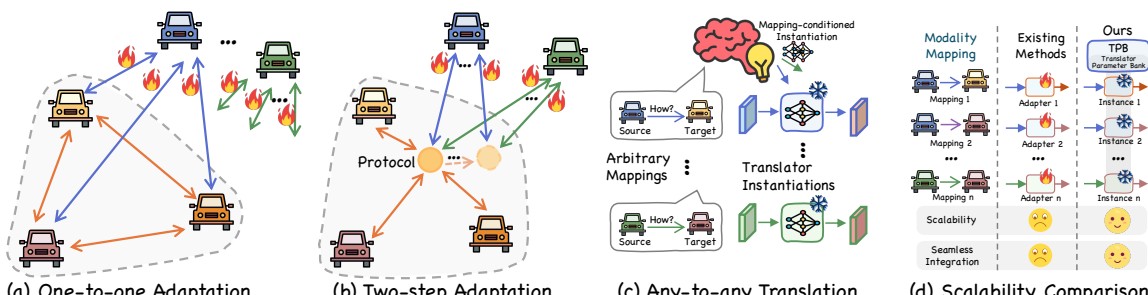

*Figure 1.* Comparison of heterogeneous collaboration perception paradigms and our proposed `UniTrans`. (a) One-to-one adaptation: trains a dedicated adapter for each pairwise modality mapping, incurring substantial training complexity. (b) Two-step adaptation: only needs to learn an adapter to the protocol space, which is not universally suitable for newly emerging agents. (c) Any-to-any translation (ours): instantiates a translator conditioned on the dynamic modality mapping, enabling any-to-any feature translation without retraining. (d) Scalability comparison: `UniTrans` achieves flexible scalability and enables seamless integration of unseen heterogeneous agents.

model and data privacy constraints. These limitations hinder seamless integration and motivate a key question: *How can we pretrain a universal model for any-to-any modality translation that directly supports newly emerging agents without any retraining or fine-tuning?*

To this end, we propose `UniTrans`, a universal any-to-any heterogeneous feature translation framework that instantiates mapping-specific translators on the fly for arbitrary modality mappings, as illustrated in Fig. 1(c). By inferring modality mappings in a modality-intrinsic latent space and instantiating translators via mapping-conditioned expert parameter combination, `UniTrans` achieves flexible scalability and enables seamless integration of unseen heterogeneous agents (Fig. 1(d)).

Specifically, we design a two-stage pretraining pipeline for `UniTrans` to enable effective on-the-fly instantiation of heterogeneous feature translators, as illustrated in Fig. 2. To enable more targeted, real-time translator instantiation from single-frame heterogeneous feature, we first pretrain a Modality-Intrinsic Encoder (MIE) that extracts modality-specific, scene-invariant intrinsic codes from intermediate features and organizes them into a modality-intrinsic latent space, where newly emerging modalities can be reliably localized and compared to estimate their mappings. Next, to handle diverse and evolving modality mappings, we decompose complex mappings into specialized experts by learning a reusable Translator Parameter Bank (TPB) and adopting a mapping-conditioned parameter combination strategy. A Modality Mapping Router (MMR) then predicts combination coefficients from the inferred mapping and instantiates a mapping-specific feature translator. Importantly, for real-time, on-board efficiency, `UniTrans` combines parameters to synthesize a single translator, rather than executing multiple experts and mixing their outputs. At inference time, all agents share the same TPB and MMR, and `UniTrans` instantiates a dedicated translator on the fly for any newly emerging heterogeneous agent to efficiently translate its intermediate features into target-consistent representations

in a zero-shot manner, eliminating repeated retraining and enabling scalable real-world collaboration.

Accordingly, the main contributions of this work can be summarized as follows:

- We propose `UniTrans`, the first universal any-to-any translation model for heterogeneous collaborative perception, which instantiates a mapping-conditioned feature translator on the fly to translate features from newly emerging agents.

- We design a tailored pipeline for `UniTrans` by pretraining a Modality-Intrinsic Encoder, a Modality Mapping Router, and a reusable Translator Parameter Bank, while learning parameter combination coefficients as a function of the modality mapping. At inference time, this enables efficient and accurate zero-shot feature translation without any additional retraining.

- We conduct extensive experiments on both simulated and real-world datasets, including OPV2V-H and DAIR-V2X. The results show that our method improves perception performance by up to 10%, enabling efficient any-to-any translation through a universal model.

## 2. Related Work

### 2.1. Heterogeneous Collaborative Perception

Collaborative perception mitigates the limited sensing range and occlusion of single agents by enabling information sharing among connected vehicles. Intermediate fusion has become the dominant paradigm due to its favorable trade-off between communication bandwidth and perception accuracy (Wang et al., 2020; Yazgan et al., 2024). However, most early pipelines assume homogeneous agents with identical sensor setups and model architectures, which limits scalability in real-world heterogeneous deployments (Yazgan et al., 2024). To relax this assumption, recent works study heterogeneous collaboration via feature translation

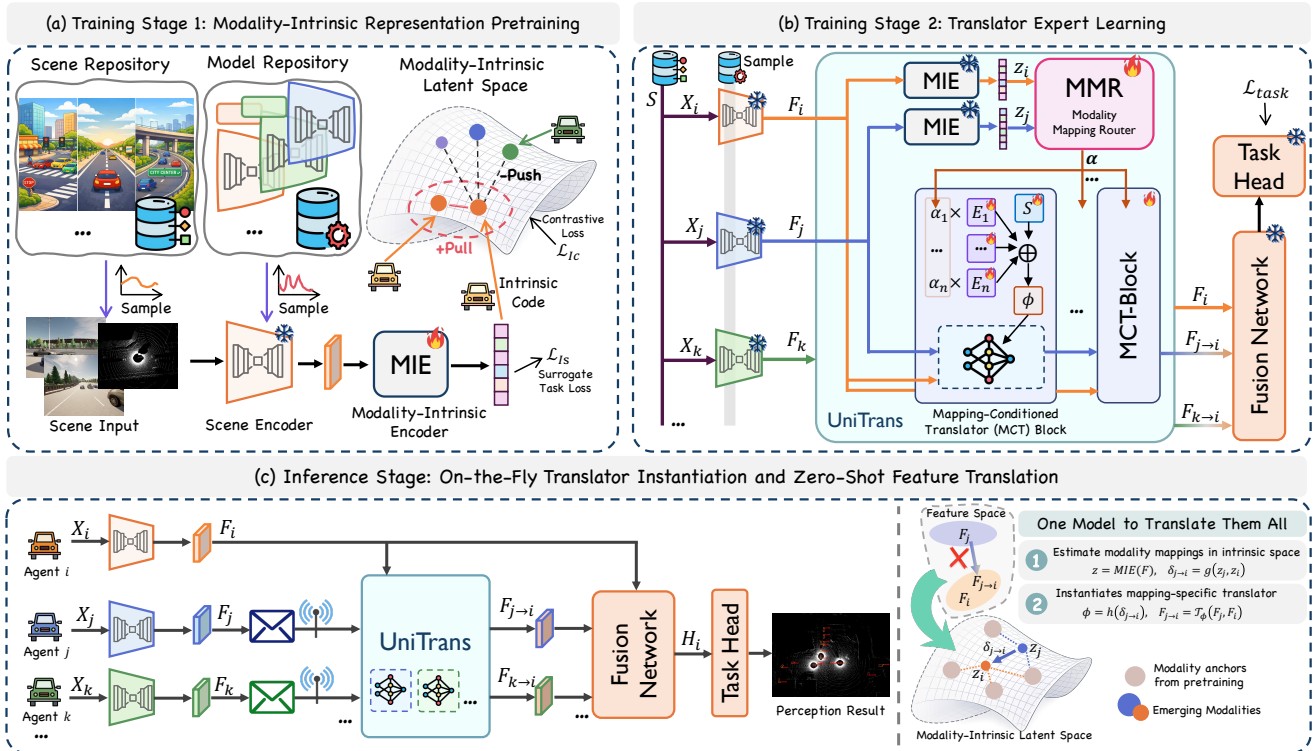

*Figure 2.* The overall architecture of our proposed `UniTrans`. After a single two-stage pretraining procedure, `UniTrans` can perform zero-shot feature translation at inference time under arbitrary heterogeneous modality mappings, without any repeated retraining.

or protocol-space alignment. One-to-one adaptation learns mapping-specific adapters, such as MPDA (Xu et al., 2023), PnPDA (Luo et al., 2025), and PolyInter (Xia et al., 2025). Two-step methods build a shared protocol space, including HEAL (Lu et al., 2024), STAMP (Gao et al., 2025), and NegoCollab (Shao et al., 2025). Despite progress, these approaches typically require retraining or fine-tuning to integrate newly emerging modalities, leaving zero-shot open-world collaboration underexplored.

### 2.2. Traditional Mixture of Experts

Mixture of Experts (MoE) increases model capacity via conditional computation, where a router activates only a small subset of experts per input, enabling scalable specialization with near-constant compute (Shazeer et al., 2017; Fedus et al., 2022). MoE has become a standard recipe for scaling Transformers (Cai et al., 2025), and recent variants further improve routing efficiency and expert utilization (Guo et al., 2024; Ablin et al., 2025). Beyond language, MoE has been adopted in multimodal and vision-centric systems, including vision-language models (Li et al., 2025; Lin et al., 2026), diffusion Transformers (Cheng et al., 2025), and autonomous-driving perception such as LiDAR representation learning and efficient 3D detection (Xu et al., 2025; Liu et al., 2025). These designs require *multi*-expert forwarding and output mixing, whereas `UniTrans` instantiates a *single* mapping-specific translator via parameter combination for more efficient inference.

## 3. Methodology

### 3.1. System Overview

**System.** As illustrated in Fig. 2(c), we consider an intermediate-fusion collaborative perception system with $N$ agents indexed by $\mathcal{A} = \{1, \dots, N\}$. Let $\mathcal{M}$ denote the set of modality types, and let $\mathcal{F}$ denote the ambient intermediate-feature space. At time $t$, each agent $i \in \mathcal{A}$ receives an observation $X_i$ (e.g., LiDAR or camera data) and extracts an intermediate feature via its encoder $f_i^{\text{enc}}$:

$$F_i = f_i^{\text{enc}}(X_i), \quad m_i \in \mathcal{M}, \ F_i \in \mathcal{F}_{m_i} \subset \mathcal{F}, \quad (1)$$

where $m_i$ denotes the modality of agent $i$, and $\mathcal{F}_{m_i}$ denotes the corresponding modality-specific feature subspace.

During collaborative perception, without loss of generality, we treat agent $i$ as the ego agent and agent $j$ as a neighbor. Neighbor $j$ transmits a shared representation:

$$\tilde{F}_j = f^{\text{share}}(F_j), \quad (2)$$

where $f^{\text{share}}(\cdot)$ denotes the communication packaging. After receiving $\tilde{F}_j$, the ego agent performs geometric alignment, including coordinate transformation and feature map

resizing, to obtain an aligned feature $\hat{F}_{j\to i}$. Nevertheless, $\hat{F}_{j\to i}$ may still be heterogeneous and out-of-domain for the ego fusion module; directly feeding it into the fusion network can lead to misinterpretation and degraded fused representations. Therefore, it is translated into an ego-consistent representation using a feature translator $\mathcal{T}_{\phi_{j\to i}}$:

$$F_{j\to i} = \mathcal{T}_{\phi_{j\to i}}\Big(\hat{F}_{j\to i}, F_i\Big), \qquad (3)$$

where $\phi_{j\to i}$ denotes the translator parameters for the modality mapping from agent $j$ to agent $i$.

The ego agent then fuses its own feature with translated neighbor features:

$$H_i = f_i^{\text{fusion}}\Big(F_i, \{F_{j\to i}\}_{j\in\mathcal{N}(i)}\Big), \qquad (4)$$

where $\mathcal{N}(i)$ is the set of neighbors communicating with ego $i$. Finally, a task head produces the perception prediction:

$$\hat{Y}_i = f_i^{\text{task}}(H_i). \qquad (5)$$

**Real-world zero-shot setting.** We consider an open-world deployment where both modalities and scenes evolve over time. Formally, we partition the modality set $\mathcal{M}$ and the scene set $\mathcal{D}$ into two disjoint subsets:

$$\mathcal{M} = \mathcal{M}_{\text{tr}} \cup \mathcal{M}_{\text{em}}, \qquad \mathcal{M}_{\text{tr}} \cap \mathcal{M}_{\text{em}} = \varnothing, \qquad (6)$$

$$\mathcal{D} = \mathcal{D}_{\text{tr}} \cup \mathcal{D}_{\text{te}}, \qquad \mathcal{D}_{\text{tr}} \cap \mathcal{D}_{\text{te}} = \varnothing, \qquad (7)$$

where $\mathcal{M}_{\text{tr}}$ contains modalities observed during training and $\mathcal{M}_{\text{em}}$ contains newly emerging modalities encountered only at inference. Similarly, $\mathcal{D}_{\text{tr}}$ denotes the training scene set, while $\mathcal{D}_{\text{te}}$ represents test-time scenes.

At inference, the ego vehicle may encounter newly emerging modalities in $\mathcal{M}_{\text{em}}$ and scene distributions from $\mathcal{D}_{\text{te}}$ that are never observed during training. Since additional training or fine-tuning is often impractical in deployment, we target *on-the-fly* heterogeneous feature translation, converting received neighbor features into ego-consistent representations to maximize downstream perception performance:

$$\forall\, m_j \in \mathcal{M}_{\text{em}},\ m_i \in \mathcal{M}, \quad \hat{F}_{j\to i} \mapsto F_{j\to i} \in \mathcal{F}_{m_i},$$
$$\text{s.t.} \quad \min \mathcal{L}_{\text{task}}(\hat{Y}_i, Y_i) \quad \text{without retraining.} \qquad (8)$$

### 3.2. Mechanism of Any-to-Any Instantiation

For a modality type $m \in \mathcal{M}$, features concentrate in a modality-specific subspace $\mathcal{F}_m \subset \mathcal{F}$. The system thus maintains a collection of modality subspaces $\{\mathcal{F}_m\}_{m\in\mathcal{M}}$. Heterogeneous feature translation from a source (neighbor) modality $m_j$ to a target (ego) modality $m_i$ can be viewed as learning an ideal subspace mapping:

$$\Delta_{j\to i} : \mathcal{F}_{m_j} \to \mathcal{F}_{m_i}, \qquad (9)$$

so that for any $F_j \in \mathcal{F}_{m_j}$, the translated feature $\Delta_{j\to i}(F_j)$ lies in $\mathcal{F}_{m_i}$ and becomes compatible with ego fusion. Most existing approaches (Xu et al., 2023; Luo et al., 2025) approximate $\Delta j \to i$ by training an adapter with many samples spanning both subspaces and optimizing its parameters via backpropagation on translation objectives (e.g., feature reconstruction losses).

In the real-world setting, a newly emerging modality induces a previously unseen subspace $\mathcal{F}_{m_j}$, while retraining is often impractical. Estimating $\Delta_{j\to i}$ from a single test-time feature instances is ill-posed because intermediate features are high-dimensional and entangle scene factors with modality factors, so sparse samples cannot characterize the geometry of $\mathcal{F}_{m_j}$.

**Extracting generalizable modality-specific intrinsic codes.** Since collaborative perception agents share the same task objective of improving the ego agent's perception, their intermediate features exhibit task-driven semantic commonality and can be regarded as lying in a common feature space $\mathcal{F}$ that admits a unified intrinsic embedding.

Rather than directly estimating the transformation $\mathcal{F}_{m_j} \to \mathcal{F}_{m_i}$ in the high-dimensional feature space $\mathcal{F}$, `UniTrans` infers modality mappings indirectly in a *modality-intrinsic latent space* $\mathcal{Z}$. Specifically, we first map intermediate features into a low-dimensional *Modality-Intrinsic Latent Space* that suppresses scene-dependent variations while preserving modality-specific structure. We learn a modality-intrinsic encoder (MIE):

$$z = \text{MIE}(F), \qquad z \in \mathbb{R}^d, \qquad (10)$$

where $z$ is the intrinsic code of feature $F$, and the training objective encourages codes from the same modality to concentrate in a compact region of the intrinsic space. We denote the intrinsic anchor of modality $m$ by $\mathcal{Z}_m \subset \mathbb{R}^d$ and formalize this property as:

$$F \in \mathcal{F}_m \ \Rightarrow\ z = \text{MIE}(F) \in \mathcal{Z}_m \subset \mathcal{Z}, \qquad (11)$$

where $\mathcal{Z}$ denotes the modality-intrinsic space. With sufficient pretraining over diverse modalities in $\mathcal{M}_{\text{tr}}$, MIE *learns a transferable encoding $\mathcal{F} \to \mathcal{Z}$ that disentangles modality cues from scene variations.* As a result, newly emerging modalities can be localized to consistent intrinsic regions in $\mathcal{Z}$ without additional training, providing the first source of zero-shot capability.

Once modalities are represented in the intrinsic space, the mapping becomes easier to characterize. We infer a mapping descriptor from intrinsic codes:

$$\delta_{j\to i} = g(z_j, z_i),\ \ z_j = \text{MIE}(F_j), z_i = \text{MIE}(F_i), \quad (12)$$

where $g(\cdot)$ summarizes the relative intrinsic geometry between source and target modalities (e.g., distance and direction in the intrinsic space). This replaces learning $\Delta_{j\to i}$

directly in $\mathcal{F}$ with learning a compact relation in the intrinsic space, which is more stable under scene shifts.

**Decomposing the mapping space with a TPB for instantiation.** Even with a compact descriptor, the space of modality mappings remains diverse in real-world deployments. Modeling all mappings with a single monolithic translator can underfit heterogeneous transformations. We therefore introduce a TPB that stores a finite set of reusable translator expert parameters. A MMR takes $(z_j, z_i)$ as input, first estimating the modality mapping $\delta_{j \to i}$ in the intrinsic space and then outputting parameter combination coefficients:

$$\boldsymbol{\alpha}_{j \to i} = \mathrm{MMR}(z_j, z_i). \qquad (13)$$

We instantiate mapping-specific translators by linearly combining TPB parameters (see Eq. (24) for the parameter combination details):

$$\phi_{j \to i} = \mathrm{Combine}(\boldsymbol{\alpha}_{j \to i};\ \mathrm{TPB}). \qquad (14)$$

Following Eq. (12)– Eq. (14), `UniTrans` learns a transferable mapping from the modality-mapping space $\mathcal{H}$ to the translator-instance space $\Phi$:

$$\delta_{j \to i} \in \mathcal{H}\ \to \phi_{j \to i} \in \Phi, \qquad (15)$$

where each $\phi_{j \to i}$ specifies one instantiated translator, and $\Phi$ contains a large family of such translator instances. By training MMR, we capture a shared regularity that *transforms arbitrary modality mappings in $\mathcal{H}$ to corresponding translator instances in $\Phi$*, enabling synthesis of mapping-specific translators for unseen modality pairs without retraining, which constitutes our second source of zero-shot capability.

**Together.** Combining feature-to-intrinsic-code generalization via MIE with mapping-to-translator generalization via translator parameter combination, `UniTrans` takes the current neighbor–ego feature pair as input, instantiates a suitable mapping-specific translator on the fly, and performs translation in Eq. (3). This enables any-to-any inference-time translation for newly emerging modalities under real-world constraints.

### 3.3. Stage 1: Modality-Intrinsic Representation Pretraining

Intermediate features entangle scene content and modality statistics, making modality-mapping estimation unreliable under shifts and for emerging modalities. We therefore pretrain an MIE to extract scene-invariant intrinsic codes for robust mapping estimation.

**Pretraining data construction.** We assume access to a large-scale scene repository $\mathcal{D}_{\mathrm{tr}}$ and a model repository $\mathcal{R}_{\mathrm{tr}}$ that covers modalities in $\mathcal{M}_{\mathrm{tr}}$. For a sampled scene $s \sim \mathcal{D}_{\mathrm{tr}}$, we assign each agent a random modality label

$m \in \mathcal{M}_{\mathrm{tr}}$ and use the corresponding encoder $f_m^{\mathrm{enc}} \in \mathcal{R}_{\mathrm{tr}}$ to extract an intermediate feature $F = f_m^{\mathrm{enc}}(X)$, and MIE then produces an intrinsic code $z = \mathrm{MIE}(F)$, where $d$ is the embedding dimension. These codes are used to construct a modality-intrinsic latent space.

**Modality-Intrinsic Encoder.** MIE emphasizes modality-dependent statistics that are relatively insensitive to scene variations. Given a BEV feature map $F \in \mathbb{R}^{C \times H \times W}$, where $C$ is the channel dimension and $H, W$ denote the spatial height and width of the featire, we compute first-order channel statistics:

$$
\begin{aligned}
\mu(F) &= \frac{1}{HW} \sum\nolimits_{h,w} F(:, h, w), \\
\sigma(F) &= \sqrt{\frac{1}{HW} \sum\nolimits_{h,w} \big(F(:, h, w) - \mu(F)\big)^2},
\end{aligned}
\qquad (16)
$$

and a second-order channel correlation (Gram) descriptor (Gatys et al., 2015) on a spatially downsampled feature $\bar{F}$:

$$
\begin{aligned}
\bar{F} &= \mathrm{Pool}(F) \in \mathbb{R}^{C \times H' \times W'}, \\
G(F) &= \frac{1}{H'W'} \bar{F}_{(C \times N)} \bar{F}_{(C \times N)}^{\top} \in \mathbb{R}^{C \times C},
\end{aligned}
\qquad (17)
$$

where $N = H'W'$ and $\bar{F}_{(C \times N)}$ reshapes $\bar{F}$ into $C \times N$. We compress $G(F)$ with an MLP projector $\psi_G(\cdot)$ and extract global response features $r(F)$ using a lightweight branch that aggregates channel- and spatial-level cues. MIE then fuses all descriptors:

$$z = \psi_I\Big([\mu(F),\ \sigma(F),\ r(F),\ \psi_G(G(F))]\Big), \qquad (18)$$

where $\psi_I(\cdot)$ is a fusion MLP.

**Training objective.** To structure the modality-intrinsic latent space, we jointly optimize a contrastive objective and a surrogate modality classification objective. Given a mini-batch of intrinsic codes $\{z_a\}$ with modality labels $\{m_a\}$, we define the positive set for anchor $a$ as $\mathcal{P}(a) = \{p \neq a \mid m_p = m_a\}$ and adopt an InfoNCE-style (Chen et al., 2020; Poole et al., 2019) contrastive loss to pull together codes from the same modality while pushing apart codes from different modalities:

$$\mathcal{L}_{\mathrm{IC}} = \sum_a \left[ -\log \frac{\sum_{p \in \mathcal{P}(a)} \exp\big(\mathrm{sim}(z_a, z_p)/\tau\big)}{\sum_{b \neq a} \exp\big(\mathrm{sim}(z_a, z_b)/\tau\big)} \right], \qquad (19)$$

where $\mathrm{sim}(\cdot, \cdot)$ denotes cosine similarity and $\tau$ is a temperature hyperparameter. To further improve modality identifiability, we train a lightweight MLP head $q(\cdot)$ to predict the modality label from $z$:

$$
\begin{aligned}
\hat{p}(m \mid z) &= \mathrm{softmax}\big(q(z)\big), \\
\mathcal{L}_{\mathrm{IS}} &= -\mathbb{E}_{(z,m)}\big[\log \hat{p}(m \mid z)\big].
\end{aligned}
\qquad (20)
$$

The Stage-1 objective is:

$$\mathcal{L}_{\text{stage1}} = \mathcal{L}_{\text{IC}} + \lambda_{\text{IS}}\mathcal{L}_{\text{IS}}, \quad (21)$$

where $\lambda$IS is a scalar balancing hyperparameter.

### 3.4. Stage 2: Translator Expert Learning

Stage 1 provides a scene-invariant modality-intrinsic space where modality relations are stable. `UniTrans` estimates modality mappings in this intrinsic space, and MMR converts each mapping into TPB combination weights to instantiate a mapping-specific feature translator.

**Translator Parameter Bank (TPB).** TPB stores $K$ reusable translator experts together with a shared expert:

$$\text{TPB} = \left(\{\Theta^{(k)}\}_{k=1}^{K}, \Theta^{(0)}\right), \quad (22)$$

where each $\Theta^{(k)}$ denotes a full set of parameters of a base translator backbone composed of stacked Mapping-Conditioned Translator (MCT) blocks, and $\Theta^{(0)}$ is a shared expert capturing mapping-invariant translation primitives. Each MCT block follows a sparse Transformer design consistent with prior collaborative perception frameworks (Xu et al., 2022a; 2023), where the instantiated parameters are applied to its linear projections.

**Modality Mapping Router (MMR).** Given intrinsic codes $z_j = \text{MIE}(F_j)$ and $z_i = \text{MIE}(F_i)$, MMR first forms a compact mapping descriptor and then predicts expert mixing coefficients:

$$\delta_{j \to i} = g(z_j, z_i), \quad \boldsymbol{\alpha}_{j \to i} = \text{softmax}\big(h(\delta_{j \to i})\big) \in \mathbb{R}^K, \quad (23)$$

where $g(\cdot)$ and $h(\cdot)$ are lightweight linear projectors, $\sum_{k=1}^{K} \alpha_{j \to i}^{(k)} = 1$, and $K$ denotes the number of translator parameter experts stored in the TPB.

**On-the-fly parameter instantiation.** We instantiate a mapping-specific translator by combining the shared expert with a weighted mixture of the $K$ translator experts' parameter:

$$\phi_{j \to i} = \Theta^{(0)} + \sum_{k=1}^{K} \alpha_{j \to i}^{(k)} \Theta^{(k)}. \quad (24)$$

The instantiated translator then performs heterogeneous feature translation in Eq. (3).

**Training objective.** To learn a reusable TPB and enable MMR to predict accurate parameter combination coefficients for each modality mapping, we optimize objectives that supervise both translation quality and routing behavior.

We first supervise the translation quality of the instantiated translator with two complementary signals: the downstream task loss $\mathcal{L}_{\text{task}}(\hat{Y}_i, Y_i)$ and a feature-level distillation loss. During training, we construct a teacher ego-domain feature by encoding the neighbor observation $X_j$ with the ego encoder, $F_{j \to i}^{\star} = f_{m_i}^{\text{enc}}(X_j)$, and minimize

$$\mathcal{L}_{\text{feat}} = \left\| F_{j \to i} - F_{j \to i}^{\star} \right\|_2^2. \quad (25)$$

To make routing consistent with modality mappings, we assign each translation pair $a$ a mapping label $\ell_a = (m_j, m_i)$ and a routing vector $\boldsymbol{\alpha}_a \in \mathbb{R}^K$. Let $\mathcal{P}(a) = \{p \neq a \mid \ell_p = \ell_a\}$ denote positives that share the same mapping label. We then apply an InfoNCE loss on routing vectors:

$$\mathcal{L}_{\text{ctr}} = \sum_a \left[ -\log \frac{\sum_{p \in \mathcal{P}(a)} \exp\big(\text{sim}(\boldsymbol{\alpha}_a, \boldsymbol{\alpha}_p)/\tau_{\alpha}\big)}{\sum_{b \neq a} \exp\big(\text{sim}(\boldsymbol{\alpha}_a, \boldsymbol{\alpha}_b)/\tau_{\alpha}\big)} \right], \quad (26)$$

where $\text{sim}(\cdot, \cdot)$ is cosine similarity and $\tau_{\alpha}$ is the temperature for routing contrast.

We further regularize the router to avoid expert collapse. For expert $k \in \{1, \ldots, K\}$, we define its importance and load over a mini-batch of $B$ translation pairs indexed by $a \in \{1, \ldots, B\}$:

$$\text{imp}_k = \sum_{a=1}^{B} \alpha_{a,k}, \text{load}_k = \sum_{a=1}^{B} \mathbb{I}[k \in \text{TopK}(\boldsymbol{\alpha}_a)], \quad (27)$$

where $\mathbb{I}(\cdot)$ is an indicator function that equals 1 if the condition holds and 0 otherwise. With router logits $u_{a,k}$, we regularize the router using a Switch-style load-balancing loss (Fedus et al., 2022) and a light logit penalty (Yang et al., 2025), combined as

$$\mathcal{L}_{\text{r}} = K \sum_{k=1}^{K} \left(\frac{\text{imp}_k}{B}\right)\left(\frac{\text{load}_k}{B}\right) + \frac{1}{B} \sum_{a=1}^{B} \left(\log \sum_{k=1}^{K} \exp(u_{a,k})\right)^2. \quad (28)$$

During optimization, gradients are propagated through the instantiated translator and the downstream task head, while only MMR and TPB are updated. Overall, Stage 2 minimizes

$$\mathcal{L}_{\text{stage2}} = \mathcal{L}_{\text{task}} + \lambda_{\text{feat}}\mathcal{L}_{\text{feat}} + \lambda_{\text{ctr}}\mathcal{L}_{\text{ctr}} + \lambda_r \mathcal{L}_r, \quad (29)$$

where $\lambda_{\text{feat}}, \lambda_{\text{ctr}}$, and $\lambda_r$ are scalar hyperparameters.

## 4. Experiments

### 4.1. Settings

To comprehensively evaluate `UniTrans` under diverse heterogeneous collaboration scenarios, we conduct experiments on two widely used benchmarks, including the simulation-based OPV2V-H (Lu et al., 2024) dataset and the real-world dataset DAIR-V2X (Yu et al., 2022). Following the definition of the real-world zero-shot heterogeneous

*Table 1.* Any-to-any translation performance on OPV2V-H. Columns correspond to the ego agent modality. Neighbor modalities are drawn from the emerging-modality set $\mathcal{M}_{\mathrm{em}}$ and are kept identical across methods. Each entry reports AP@0.5/AP@0.7.

| Method | m7 | m13 | m17 | m25 | m27 | m30 | Avg. |
|---|---|---|---|---|---|---|---|
| MPDA | 0.812/0.700 | 0.696/0.600 | 0.784/0.713 | 0.767/0.692 | 0.303/0.145 | 0.368/0.197 | 0.622/0.508 |
| PnPDA | 0.798/0.713 | 0.772/0.686 | 0.744/0.688 | 0.778/0.710 | 0.416/0.192 | 0.316/0.154 | 0.637/0.524 |
| PolyInter | 0.782/0.717 | 0.685/0.645 | 0.833/0.769 | 0.788/0.728 | 0.431/0.209 | 0.342/0.165 | 0.644/0.539 |
| STAMP | 0.786/0.687 | 0.772/0.691 | 0.801/0.716 | 0.745/0.678 | 0.447/0.207 | 0.368/0.184 | 0.653/0.527 |
| NegoCollab | 0.791/0.726 | 0.789/0.707 | 0.797/0.706 | 0.773/0.696 | 0.468/0.206 | 0.355/0.188 | 0.662/0.538 |
| ConvNeXt | 0.735/0.645 | 0.708/0.673 | 0.743/0.671 | 0.741/0.678 | 0.235/0.131 | 0.236/0.130 | 0.566/0.488 |
| Transformer | 0.751/0.678 | 0.678/0.614 | 0.768/0.697 | 0.738/0.667 | 0.391/0.196 | 0.326/0.162 | 0.609/0.502 |
| Classic MoE | 0.796/0.736 | 0.774/0.719 | 0.791/0.713 | 0.758/0.705 | 0.450/0.207 | 0.348/0.184 | 0.653/0.544 |
| **UniTrans** | **0.835/0.778** | **0.843/0.799** | **0.862/0.806** | **0.850/0.803** | **0.497/0.243** | **0.406/0.202** | **0.716/0.605** |

*Table 2.* Any-to-any translation performance on DAIR-V2X under different ego modalities. Each entry reports AP@0.5/AP@0.7. Columns indicate the ego modality; neighbor modalities are drawn from the emerging-modality set $\mathcal{M}_{\mathrm{em}}$.

| Method | m7 | m13 | m17 | m25 | m27 | m30 | Avg. |
|---|---|---|---|---|---|---|---|
| MPDA | 0.650/0.563 | 0.646/0.555 | 0.654/0.557 | 0.619/0.496 | 0.211/0.051 | 0.157/0.021 | 0.489/0.374 |
| PnPDA | 0.661/0.571 | 0.669/0.565 | 0.649/0.502 | 0.644/0.529 | 0.210/0.071 | 0.216/0.072 | 0.508/0.385 |
| PolyInter | 0.640/0.554 | 0.635/0.542 | 0.653/0.546 | 0.655/0.532 | 0.168/0.031 | 0.166/0.031 | 0.486/0.373 |
| STAMP | 0.647/0.550 | 0.645/0.549 | 0.650/0.553 | 0.658/0.534 | 0.210/0.064 | 0.206/0.051 | 0.503/0.384 |
| NegoCollab | 0.662/0.570 | 0.650/0.555 | 0.653/0.556 | 0.642/0.538 | 0.225/0.061 | 0.216/0.053 | 0.509/0.389 |
| ConvNeXt | 0.628/0.535 | 0.645/0.549 | 0.606/0.496 | 0.606/0.518 | 0.110/0.020 | 0.119/0.021 | 0.452/0.357 |
| Transformer | 0.652/0.544 | 0.641/0.545 | 0.642/0.530 | 0.628/0.513 | 0.192/0.031 | 0.152/0.042 | 0.484/0.367 |
| Classic MoE | 0.684/0.565 | 0.661/0.558 | 0.660/0.531 | 0.663/0.525 | 0.233/0.091 | 0.210/0.051 | 0.523/0.388 |
| **UniTrans** | **0.709/0.597** | **0.692/0.573** | **0.693/0.579** | **0.688/0.551** | **0.252/0.115** | **0.235/0.092** | **0.553/0.421** |

collaborative perception setting in Sec. 3.1, we consider a representative set of LiDAR and camera encoders, including PointPillars (Lang et al., 2019), SECOND (Yan et al., 2018), VoxelNet (Zhou & Tuzel, 2018), and Lift-Splat-Shoot (Philion & Fidler, 2020). By varying backbone architectures and network depths, we construct 30 modality categories (details in Appendix A.3). Following Eq. (6), we partition modalities by holding out six emerging modalities, $\mathcal{M}_{\mathrm{em}} = \{\mathrm{m7, m13, m17, m25, m27, m30}\}$, which are only encountered at inference time, while the remaining modalities are used for training.

All modality-specific perception backbones are pretrained beforehand and shared across all baselines. To ensure a fair comparison, all heterogeneous feature translation methods are trained on the same $\mathcal{M}_{\mathrm{tr}}$ and $\mathcal{D}_{\mathrm{tr}}$ for 70 epochs using the

same learning rate. Since baseline translators are not inherently designed to accommodate numerous newly emerging modalities, we further strengthen their training protocol to better accommodate newly emerging modalities to a certain extent (see Sec. A.4), making the comparison both fair and informative.

### 4.2. Performance Comparison

Tab. 1 and Tab. 2 evaluate any-to-any feature translation by fixing the ego modality per column and sampling neighbor modalities from the emerging-modality set $\mathcal{M}_{\mathrm{em}}$. We compare against representative heterogeneous collaboration baselines, including one-to-one adaptation methods (MPDA (Xu et al., 2023), PnPDA (Luo et al., 2025),

PolyInter (Xia et al., 2025)) and protocol-based methods (STAMP (Gao et al., 2025), NegoCollab (Shao et al., 2025)). To further contextualize performance, we additionally include common one-to-one adapter backbones (ConvNeXt (Liu et al., 2022; Gao et al., 2025) and Transformer (Xu et al., 2022a)), as well as a Classic MoE (Guo et al., 2024; Xu et al., 2025) baseline that routes experts directly from ego–neighbor features.

As shown in Tab. 1, UniTrans achieves the best average accuracy across ego modalities, reaching 0.716/0.605 (AP@0.5/AP@0.7), outperforming the strongest baseline NegoCollab (0.662/0.538) and Classic MoE (0.653/0.544). Notably, UniTrans consistently improves performance for newly emerging LiDAR ego modalities (m7–m25), indicating that learning mapping-conditioned parameter combination yields robust translation across diverse LiDAR-target feature spaces. When the ego modality switches to camera features (m27, m30), all methods degrade substantially because the ego camera has a limited field of view and the collaboration must bridge a pronounced cross-modality gap between LiDAR and camera intermediate representations; nevertheless, UniTrans remains the top performer (0.497/0.243 on m27 and 0.406/0.202 on m30). In the Camera-target translation setting, the underlying modality mapping becomes highly complex due to the severe LiDAR-Camera representation gap. By projecting features into the modality-intrinsic space, UniTrans can measure modality relations more reliably and instantiate a dedicated translator via TPB parameter combination, effectively resolving this challenging heterogeneous translation with a mapping-specialized instance. The Classic MoE results further suggest that routing directly in the high-dimensional feature space makes it difficult to derive modality-mapping-aware expert selection and output aggregation.

Tab. 2 shows a similar trend in the real-world DAIR-V2X setting, where distribution shifts make translation more challenging overall. UniTrans attains the highest average score of 0.553/0.421, improving over the best prior method NegoCollab (0.509/0.389) and Classic MoE (0.523/0.388). Under the more complex modality mappings encountered in real-world scenarios, UniTrans still maintains strong perception performance, corroborating improved cross-modality generalization under domain shifts enabled by its mapping-conditioned translator instantiation mechanism.

Tab. 3 profiles the translator on the OPV2V-H test set. UniTrans instantiates a single translator via linear TPB parameter combination, enabling one-pass inference with low cost (109.3 GFLOPs; 6.865 ms CPU / 53.760 ms CUDA). By contrast, Classic MoE executes multiple experts (TopK=3 in our setting) and mixes their outputs, leading to much higher overhead. Overall, UniTrans is more efficient while achieving stronger translation performance,

*Table 3.* Profiling results on the test set, reporting the average per-scene GFLOPs, CPU time, and CUDA time.

| Method | GFLOPs | CPU (ms) | CUDA (ms) |
|---|---|---|---|
| MPDA | 124.6 | 11.852 | 46.814 |
| Classic MoE | 245.5 | 89.078 | 141.352 |
| UniTrans | 109.3 | 6.865 | 53.760 |

*Table 4.* Component ablations on OPV2V-H.

| ID | $\mathcal{L}_{\text{IC}}$ | $\mathcal{L}_{\text{IS}}$ | $\mathcal{L}_{\text{task}}$ | $\mathcal{L}_{\text{feat}}$ | $\mathcal{L}_{\text{ctr}}$ | $\mathcal{L}_r$ | AP@0.5 | AP@0.7 |
|---|---|---|---|---|---|---|---|---|
| 0 | ✓ | ✓ | ✓ | ✓ | ✓ | ✓ | 0.716 | 0.605 |
| 1 | ✗ | ✓ | ✓ | ✓ | ✓ | ✓ | 0.685 | 0.575 |
| 2 | ✓ | ✗ | ✓ | ✓ | ✓ | ✓ | 0.694 | 0.583 |
| 3 | ✗ | ✗ | ✓ | ✓ | ✓ | ✓ | 0.662 | 0.540 |
| 4 | ✓ | ✓ | ✗ | ✓ | ✓ | ✓ | 0.691 | 0.579 |
| 5 | ✓ | ✓ | ✓ | ✗ | ✓ | ✓ | 0.653 | 0.531 |
| 6 | ✓ | ✓ | ✓ | ✓ | ✗ | ✓ | 0.688 | 0.576 |
| 7 | ✓ | ✓ | ✓ | ✓ | ✓ | ✗ | 0.708 | 0.598 |

making it better suited for real-world, on-vehicle collaborative perception.

### 4.3. Ablation Studies

Tab. 4 reports component ablations on OPV2V-H by removing one objective at a time from our full training recipe. The complete model (ID 0) achieves the best performance, validating the effectiveness of jointly optimizing Stage 1 intrinsic-space representation and Stage 2 mapping-conditioned translator instantiation. When removing $\mathcal{L}_{\text{IC}}$ or $\mathcal{L}_{\text{IS}}$ individually (ID 1–2), performance drops to 0.685/0.575 and 0.694/0.583, respectively, indicating that both contrastive clustering and modality identifiability are necessary to produce a well-behaved modality-intrinsic space. A pronounced degradation is observed when both $\mathcal{L}_{\text{IC}}$ and $\mathcal{L}_{\text{IS}}$ are removed (ID 3, 0.662/0.540), which effectively disables Stage 1 training, and the MMR must then infer routing directly from current intermediate features without intrinsic regularization, making routing harder and reducing generalization to unseen modality mappings. Stage 2 objectives are also consistently beneficial. Removing the downstream task loss $\mathcal{L}_{\text{task}}$ (ID 4) reduces performance to 0.691/0.579, indicating that task-level supervision is necessary to retain task-relevant semantics. Dropping the feature distillation term $\mathcal{L}_{\text{feat}}$ (ID 5) causes the largest Stage 2 degradation (0.653/0.531), highlighting its role as a strong, stable teacher signal for aligning translated features to the ego domain, especially under large modality gaps. Removing the MMR routing regularizers (i.e., $\mathcal{L}_{\text{ctr}}$ and $\mathcal{L}_r$, ID 6, 7) consistently hurts performance, suggesting they enforce mapping-consistent routing and improve MMR's ability to generalize when synthesizing translator instances across diverse modality pairs. Overall, the ablations confirm that both Stage 1 intrinsic-space learning and Stage 2 mapping-conditioned instantiation are essential, with each

component contributing materially to the final any-to-any translation accuracy.

## 5. Conclusion

In this paper, we propose `UniTrans`, a universal any-to-any feature modality translation framework that instantiates mapping-specific translators on the fly, addressing a key deployment bottleneck in collaborative perception where repeated cross-agent joint training is often impractical. With a single pretrained universal model, `UniTrans` supports seamless collaboration with newly emerging heterogeneous agents by inferring the modality mapping, instantiating a mapping-conditioned translator on the fly, and translating their intermediate features into target-consistent representations, with strong generalization to unseen counterparts. Future work will explore more expressive yet efficient instantiation strategies and further scale the pretrained foundations. We hope `UniTrans` advances open-world zero-shot heterogeneous feature translation and accelerates the deployment of efficient and reliable autonomous systems.

## Impact Statement

This paper studies open-world, zero-shot translation of heterogeneous intermediate features for collaborative perception, aiming to reduce the need for repeated cross-agent retraining and to improve scalability in practical deployments. The anticipated positive impact is enabling more reliable and efficient information sharing among connected agents, which may support safer and more capable perception in autonomous driving and related robotics systems. Potential risks include misuse in safety-critical settings without sufficient validation, distribution shifts that degrade translation quality, and privacy considerations when intermediate representations are exchanged across entities. We mitigate these concerns by focusing on feature-level translation without requiring access to raw sensor data, and by emphasizing controlled evaluation; nonetheless, real-world deployment should incorporate rigorous safety testing, monitoring, and system-level safeguards.

## Acknowledgements

This work was supported in part by the National Key Research and Development Program of China under Grant 2023YFB4301900, in part by the Natural Science Foundation of China under Grant 62272053 and Grant 62472048, in part by the Beijing Nova Program under Grant 20230484364, in part by Beijing Natural Science Foundation under Grant L242081, and in part by BUPT Excellent Ph.D. Students Foundation.

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

# A. EXPERIMENTAL SETTINGS

## A.1. Datasets

**OPV2V-H** (Lu et al., 2024) is a simulation benchmark specifically tailored for heterogeneous collaborative perception research. Built upon the OpenCDA framework (Xu et al., 2021) and CARLA simulator, it extends the original OPV2V dataset by introducing diverse sensor configurations to simulate realistic hardware discrepancies. Unlike standard homogenous settings, agents in OPV2V-H are equipped with varying LiDAR resolutions (16-, 32-, and 64-channels) alongside RGB and depth cameras. The dataset features dynamic traffic scenes with 2 to 7 interacting agents per frame, serving as our primary testbed for assessing immutable heterogeneity.

**DAIR-V2X** (Yu et al., 2022) represents the first large-scale real-world collaborative perception dataset for Vehicle-Infrastructure Cooperation (VIC). It contains 9,000 frames of synchronized data captured from a real-world vehicle and a Roadside Unit (RSU). A critical challenge in this dataset is the extreme sensor heterogeneity: the RSU is equipped with a high-resolution 300-channel LiDAR, whereas the vehicle employs a 40-channel LiDAR. We use DAIR-V2X to validate the generalization capability of GPA in real-world environments with significant domain gaps.

## A.2. Implementations

The proposed models are implemented in PyTorch and trained on NVIDIA RTX 4090 GPUs using the Adam optimizer (Kingma & Ba, 2015) with a learning rate of $1 \times 10^{-3}$. For the task loss, we adopt standard 3D detection objectives, using Smooth-$L_1$ loss for box regression and focal loss for classification, consistent with prior baselines (Hu et al., 2022; Xu et al., 2022b). The remaining hyperparameters are summarized in Tab. 5. Experimental setups follow the perception range specifications in Tab. 6. During training, models are optimized on the training split and we select the checkpoint with the lowest validation loss; final results are reported on the test split. All settings are kept identical across competing methods for fair comparison.

## A.3. Modality Setting

Table 8 summarizes the heterogeneous modality set used in our experiments, covering both LiDAR- and camera-based perception stacks with diversified encoder architectures and model capacities. For LiDAR, we vary voxelization granularity and backbone capacity to induce systematic modality shifts, where *Normal/Medium/Large* correspond to three parameter-scaled variants under a fixed voxel size, and PointPillar/SECOND/VoxelNet are instantiated with multiple voxel resolutions to further diversify the sensing-to-feature formation process. For camera, we adopt the LSS

pipeline with different image backbones (EfficientNet and ResNet families) to create complementary modality groups with distinct parameter budgets and representation biases. All perception backbones follow the same architecture and configuration as prior works (Lu et al., 2024; Gao et al., 2025), using a pyramid feature fusion network together with the standard detection head for cooperative 3D object detection.

Following the real-world zero-shot heterogeneous collaborative perception setting defined in Sec. 3.1, we partition the full modality set into *training-visible* and *inference-visible* modalities to emulate deployment-time encounters with unseen agents. Specifically, the inference-visible modalities are m7, m13, m17, m25, m27, m30, while all remaining modalities in Table 8 are used only during training. This split ensures that evaluation strictly measures real-world generalization, as the model must accommodate emerging modality instantiations at inference without additional retraining.

## A.4. Experimental Details

Following the real-world heterogeneous collaborative perception setup in Sec. 3.1, we first prepare a catalog of modality-specific, *homogeneous* perception models according to Tab. 8. These perception backbones are pretrained and then fixed, serving as the feature extractors for both our method and all baselines. We subsequently train only the heterogeneous feature *translator* components to enable collaboration across arbitrary modality pairs.

A key deployment constraint in real-world settings (Sec. 3.1) is that newly emerging agents are typically unavailable for joint retraining with existing agents. However, most prior approaches rely on mapping-specific adapter training or modality-specific fine-tuning for each new modality; without such retraining, directly translating unseen heterogeneous features often yields very poor performance.

As shown in Table 7, when evaluated over the inference modality set $\mathcal{M}_{em}$, these baselines generalize poorly to unseen modality pairs. These methods are trained for specific modality settings rather than for the zero-shot any-to-any setting considered in this paper. While such a protocol is natural for their original designs, it does not support deployment-time generalization to newly emerging modalities. Therefore, directly evaluating these methods in our open-world zero-shot setting is inherently unfavorable to them and also highlights their limited comparability to Uni-Trans.

To enable a meaningful zero-shot evaluation, we equip representative baselines with a unified pretraining protocol that minimally modifies their original architectures while allowing them to observe diverse modality pairs during train-

*Table 5.* Hyperparameters used in our experiments.

| Symbol | Value | Description |
|--------|-------|-------------|
| $\lambda_{\mathrm{IS}}$ | 0.1 | Balances $\mathcal{L}_{\mathrm{IS}}$ in Stage-1 objective (Eq. 21) |
| $\lambda_{\mathrm{feat}}$ | 5 | Weight of feature distillation loss $\mathcal{L}_{\mathrm{feat}}$ (Eq. 29) |
| $\lambda_{\mathrm{ctr}}$ | 0.01 | Weight of routing contrastive loss $\mathcal{L}_{\mathrm{ctr}}$ (Eq. 29) |
| $\lambda_r$ | 0.001 | Weight of router regularization $\mathcal{L}_z$ (Eq. 29) |
| $\tau$ | 0.9 | Temperature for contrastive learning (InfoNCE) |
| $\tau_\alpha$ | 0.9 | Temperature for routing-vector contrast (Eq. 26) |

*Table 6.* CAV perception ranges used for training and testing.

| Dataset | Training CAV range | Testing CAV range |
|---------|--------------------|--------------------|
| OPV2V-H | $[-51.2,\ -51.2,\ -3,\ 51.2,\ 51.2,\ 1]$ | $[-102.4,\ -51.2,\ -3,\ 102.4,\ 51.2,\ 1]$ |
| DAIR-V2X | $[-102.4,\ -51.2,\ -3.5,\ 102.4,\ 51.2,\ 1.5]$ | $[-102.4,\ -51.2,\ -3.5,\ 102.4,\ 51.2,\ 1.5]$ |

*Table 7.* Performance of prior methods trained under their native assumptions and evaluated in our zero-shot any-to-any setting. Each entry reports the average AP@0.5/AP@0.7 over the inference modality set $\mathcal{M}_{\mathrm{em}}$.

| Method | OPV2V | DAIR-V2X |
|--------|-------|----------|
| MPDA | 0.393 / 0.322 | 0.367 / 0.289 |
| PnPDA | 0.388 / 0.310 | 0.355 / 0.291 |
| PolyInter | 0.391 / 0.330 | 0.370 / 0.278 |
| STAMP | 0.401 / 0.334 | 0.389 / 0.307 |
| NegoCollab | 0.431 / 0.351 | 0.401 / 0.322 |
| UniTrans | **0.716 / 0.605** | **0.553 / 0.421** |

ing. Concretely, all baseline translators are trained on the same training modality set $\mathcal{M}_{\mathrm{tr}}$ and data $\mathcal{D}_{\mathrm{tr}}$ with the same training schedule as in Fig. 2(b), and are supervised by the downstream task loss and feature-level distillation when applicable. This produces a *single* pretrained translator per baseline that can be directly applied to arbitrary modality pairs at inference time.

**One-to-one adapters (MPDA, PnPDA, ConvNeXt, Transformer).** MPDA, PnPDA, and Transformer-based adapters share an attention-style formulation that can take both neighbor and ego features as inputs and translate neighbor features to be ego-consistent by conditioning on the ego feature space. We train these adapters using arbitrary modality pairs sampled from $\mathcal{D}_{\mathrm{tr}}$, with teacher feature distillation and task supervision, so that a *single* pretrained adapter can handle any-to-any translation at test time. ConvNeXt, in contrast, is commonly used as a single-input adapter and is typically trained per neighbor modality. To adapt it to any-to-any translation without changing its backbone, we feed both neighbor and ego features by concatenation at the input, en-

couraging the adapter to translate neighbor features toward the ego style under the same supervision.

**PolyInter.** Beyond a shared translation module, PolyInter introduces learnable *specific prompts* to capture modality-dependent characteristics. We thus train a universal set of prompts jointly with the translator on diverse modality pairs, enabling prompt-conditioned translation with broader any-to-any coverage.

**Protocol-space methods (STAMP, NegoCollab).** STAMP employs an *Adapter* that maps input features into a protocol space and a *Reverter* that maps protocol features into the ego space. In its original setting, a new modality requires training a modality-specific Adapter–Reverter pair, which is incompatible with the open-world zero-shot constraint. In our any-to-any setting, training a universal Adapter that maps arbitrary modality features into a shared protocol space is feasible (translate the neighbor features into protocol features). The main difficulty lies in the Reverter: mapping a single protocol feature to many possible ego domains is ill-posed if the Reverter only takes protocol features as input. To alleviate this, we modify the Reverter to be ego-conditioned by concatenating the protocol feature with the ego feature, so it can generate ego-consistent outputs conditioned on the target domain. NegoCollab follows a similar training pipeline, but its protocol space is obtained via cross-modality negotiation over all training modalities; we then supervise the Adapter to map arbitrary modality features into this negotiated protocol space.

**Classic MoE.** For the Classic MoE baseline, we use the same number of experts as `UniTrans` (eight experts). Its router takes the neighbor and ego features as input and outputs the mixing weights. During inference, each sample is forwarded through the TopK experts (TopK$= 3$ in our

setting), and their outputs are then aggregated by weighted mixing.

**Discussion.** Despite the above adaptations, these baselines still infer modality relations and perform translation directly in the original high-dimensional feature spaces, which makes mapping estimation brittle and limits their ability to handle complex modality relations. In contrast, `UniTrans` explicitly measures modality mappings in a compact modality-intrinsic space and converts them into mapping-conditioned TPB parameter compositions, instantiating a dedicated translator for each inferred modality mapping. This mechanism provides more targeted translation and stronger generalization, enabling truly any-to-any heterogeneous feature translation in open-world collaborative perception.

# B. More Details of `UniTrans`

## B.1. Stage 1: Modality-Intrinsic Representation Pretraining

Alg. 1 outlines the pretraining procedure of the Modality-Intrinsic Encoder (MIE) on the open-world training resources. We assume access to a scene repository $\mathcal{D}_{\mathrm{tr}}$ and a model repository $\mathcal{R}_{\mathrm{tr}}$ that covers the training-visible modality set $\mathcal{M}_{\mathrm{tr}}$. The repository $\mathcal{R}_{\mathrm{tr}}$ contains a diverse collection of perception encoders instantiated from representative LiDAR and camera backbones. The scene encoders are pretrained and kept fixed during Stage-1, so MIE learns from a stable and consistent set of modality-conditioned feature generators.

In each iteration, we first sample a mini-batch of scenes $\{s_b\}_{b=1}^B \sim \mathcal{D}_{\mathrm{tr}}$ and initialize an empty set $\mathcal{B}$ to collect intrinsic codes. For every agent $i$ in each sampled scene $s_b$, we draw a modality label $m_{b,i} \sim \pi(\mathcal{M}_{\mathrm{tr}})$ and assign the corresponding encoder $f_{m_{b,i}}^{\mathrm{enc}} \in \mathcal{R}_{\mathrm{tr}}$. We then encode the agent observation $X_{b,i}$ into an intermediate BEV feature map $F_{b,i} = f_{m_{b,i}}^{\mathrm{enc}}(X_{b,i})$, and feed it into MIE to obtain an intrinsic code $z_{b,i} = \mathrm{MIE}(F_{b,i})$. To encourage modality identifiability, a lightweight surrogate head $q(\cdot)$ predicts modality posteriors $\hat{p}_{b,i} = \mathrm{softmax}(q(z_{b,i}))$, and we append each pair $(z_{b,i}, m_{b,i})$ to $\mathcal{B}$. Distribution $\pi$ can be configured to balance the sampling frequency across sensor families; in our implementation, we use a weighted sampling scheme to keep the effective training frequencies of LiDAR and camera modalities comparable, improving the generality of the learned intrinsic space.

After collecting codes from all agents in the mini-batch, we construct positive sets $\mathcal{P}(a)$ within $\mathcal{B}$ using modality labels, so that codes from the same modality form positives and codes from different modalities serve as negatives. We then optimize MIE by jointly minimizing an InfoNCE-style contrastive loss $\mathcal{L}_{\mathrm{IC}}$ (Eq. (19)) and a surrogate modality

---

**Algorithm 1** Pretraining of Modality-Intrinsic Encoder

**Require:** Scene repository $\mathcal{D}_{\mathrm{tr}}$; model repository $\mathcal{R}_{\mathrm{tr}}$ covering $\mathcal{M}_{\mathrm{tr}}$; batch size $B$.
1: **while** not converged **do**
2:     Sample a mini-batch of scenes $\{s_b\}_{b=1}^B \sim \mathcal{D}_{\mathrm{tr}}$
3:     Initialize batch set $\mathcal{B} \leftarrow \emptyset$
4:     **for** each scene $s_b$ **do**
5:         **for** each agent $i$ in $s_b$ **do**
6:             Sample modality $m_{b,i} \sim \pi(\mathcal{M}_{\mathrm{tr}})$
7:             Assign encoder $f_{m_{b,i}}^{\mathrm{enc}} \in \mathcal{R}_{\mathrm{tr}}$
8:             $F_{b,i} \leftarrow f_{m_{b,i}}^{\mathrm{enc}}(X_{b,i})$
9:             $z_{b,i} \leftarrow \mathrm{MIE}(F_{b,i})$
10:           $\hat{p}_{b,i} \leftarrow \mathrm{softmax}(q(z_{b,i}))$
11:           Add $(z_{b,i}, m_{b,i})$ to $\mathcal{B}$
12:         **end for**
13:     **end for**
14:     Construct positive sets $\mathcal{P}(a)$ in $\mathcal{B}$ using modality labels
15:     Compute $\mathcal{L}_{\mathrm{IC}}$ by Eq. (19) and $\mathcal{L}_{\mathrm{S}}$ by Eq. (20)
16:     Update parameters by minimizing $\mathcal{L}_{\mathrm{stage1}}$
17: **end while**

---

classification loss $\mathcal{L}_{\mathrm{IS}}$ (Eq. (20)), with the overall objective $\mathcal{L}_{\mathrm{stage1}}$ given in Eq. (21). This training encourages MIE to produce codes that are compact and stable under scene variations, yet separable across modalities, thereby providing a structured modality-intrinsic space for Stage-2 translator instantiation.

**Discussion: vs. conventional MoE.** Conventional MoE typically mixes expert *activations* and must execute multiple experts per input, which is compute-intensive. In contrast, our instantiation combines expert *parameters* once per modality mapping (Eqs. (23)–(24)), then runs a single instantiated translator, yielding lower inference overhead while retaining the ability to represent diverse mappings through compositional experts.

## B.2. Stage 2: Translator Expert Learning

**Structure of MCT Block.** Our translator instantiation further leverages multiple Mapping-Conditioned Translator (MCT) blocks implemented as a sparse cross-attention Transformer operating on BEV features. Given ego and neighbor feature maps $F_i, F_j \in \mathbb{R}^{C \times H \times W}$, we first partition them into non-overlapping $w \times w$ windows and flatten each window into token sequences, yielding $\mathbf{T}_j, \mathbf{T}_i \in \mathbb{R}^{B \times n_W \times N \times C}$ with $N = w^2$ tokens per window and $n_W = \frac{HW}{w^2}$ windows. Within each window, the block performs multi-head *cross-attention* where queries come from neighbor tokens and keys/values come from ego tokens. Concretely, for head $h$, we compute $\mathbf{Q}^{(h)} = \mathbf{T}_j \mathbf{W}_Q^{(h)}$, $\mathbf{K}^{(h)} = \mathbf{T}_i \mathbf{W}_K^{(h)}$, $\mathbf{V}^{(h)} = \mathbf{T}_i \mathbf{W}_V^{(h)}$ with $\mathbf{W}_Q^{(h)}, \mathbf{W}_K^{(h)}, \mathbf{W}_V^{(h)} \in$

*Table 8.* Modality catalog used in heterogeneous collaborative perception. Each modality ID specifies the sensing type and encoder configuration. For LiDAR modalities, we additionally report the voxel size and the capacity variant (*Normal/Medium/Large*). Encoder parameter counts are in millions (M).

| No. | Modality ID | Sensor | Encoder / Backbone | Voxel size (m) | Size | Enc. Param. |
|---|---|---|---|---|---|---|
| 1 | m1 | Camera | LSS (EfficientNet-B0) | – | – | 14.903 M |
| 2 | m2 | LiDAR | SECOND | $[0.1, 0.1, 0.1]$ | Normal | 0.967 M |
| 3 | m3 | Camera | LSS (ResNet-101) | – | – | 1.802 M |
| 4 | m4 | LiDAR | PointPillar | $[0.4, 0.4, 0.4]$ | Normal | 0.227 M |
| 5 | m5 | LiDAR | VoxelNet | $[0.4, 0.4, 0.4]$ | Normal | 0.600 M |
| 6 | m6 | LiDAR | PointPillar | $[0.4, 0.4, 0.4]$ | Medium | 0.231 M |
| 7 | m7 | LiDAR | PointPillar | $[0.4, 0.4, 0.4]$ | Large | 0.233 M |
| 8 | m8 | LiDAR | SECOND | $[0.1, 0.1, 0.1]$ | Medium | 1.223 M |
| 9 | m9 | LiDAR | SECOND | $[0.1, 0.1, 0.1]$ | Large | 1.479 M |
| 10 | m10 | LiDAR | VoxelNet | $[0.4, 0.4, 0.4]$ | Medium | 0.711 M |
| 11 | m11 | LiDAR | VoxelNet | $[0.4, 0.4, 0.4]$ | Large | 0.822 M |
| 12 | m12 | LiDAR | PointPillar | $[0.8, 0.8, 0.8]$ | Normal | 0.223 M |
| 13 | m13 | LiDAR | PointPillar | $[0.8, 0.8, 0.8]$ | Medium | 0.227 M |
| 14 | m14 | LiDAR | PointPillar | $[0.8, 0.8, 0.8]$ | Large | 0.229 M |
| 15 | m15 | LiDAR | SECOND | $[0.2, 0.2, 0.2]$ | Normal | 0.856 M |
| 16 | m16 | LiDAR | SECOND | $[0.2, 0.2, 0.2]$ | Medium | 1.112 M |
| 17 | m17 | LiDAR | SECOND | $[0.2, 0.2, 0.2]$ | Large | 1.368 M |
| 18 | m18 | LiDAR | PointPillar | $[0.2, 0.2, 0.2]$ | Normal | 0.227 M |
| 19 | m19 | LiDAR | PointPillar | $[0.2, 0.2, 0.2]$ | Medium | 0.231 M |
| 20 | m20 | LiDAR | PointPillar | $[0.2, 0.2, 0.2]$ | Large | 0.233 M |
| 21 | m21 | LiDAR | SECOND | $[0.4, 0.4, 0.4]$ | Normal | 0.856 M |
| 22 | m22 | LiDAR | SECOND | $[0.4, 0.4, 0.4]$ | Medium | 1.112 M |
| 23 | m23 | LiDAR | SECOND | $[0.4, 0.4, 0.4]$ | Large | 1.368 M |
| 24 | m24 | LiDAR | VoxelNet | $[0.8, 0.8, 0.8]$ | Normal | 0.600 M |
| 25 | m25 | LiDAR | VoxelNet | $[0.8, 0.8, 0.8]$ | Medium | 0.711 M |
| 26 | m26 | LiDAR | VoxelNet | $[0.8, 0.8, 0.8]$ | Large | 0.822 M |
| 27 | m27 | Camera | LSS (EfficientNet-B1) | – | – | 17.409 M |
| 28 | m28 | Camera | LSS (ResNet-34) | – | – | 1.638 M |
| 29 | m29 | Camera | LSS (EfficientNet-B2) | – | – | 18.946 M |
| 30 | m30 | Camera | LSS (ResNet-50) | – | – | 1.802 M |

$\mathbb{R}^{C \times d}$ and scaling factor $1/\sqrt{d}$. The window-wise attention weights and outputs are

$$\mathbf{A}^{(h)} = \text{softmax}\left(\frac{\mathbf{Q}^{(h)}\mathbf{K}^{(h)\top}}{\sqrt{d}}\right), \quad \mathbf{O}^{(h)} = \mathbf{A}^{(h)}\mathbf{V}^{(h)}, \tag{30}$$

and the multi-head result is obtained by concatenation and a linear projection, $\mathbf{O} = \text{Concat}_h(\mathbf{O}^{(h)})\mathbf{W}_O$. This local window attention realizes *sparse* computation by restricting interactions to $w \times w$ neighborhoods, improving efficiency while preserving fine-grained spatial alignment.

Following a pre-norm residual design, the block applies LayerNorm before each sublayer and adds the cross-attention output back to the neighbor tokens. It then updates tokens via an feed-forward sublayer that maintains a Translator Parameter Bank (TPB). Given the combination coefficients predicted by MMR for the current modality mapping, the TPB parameters are linearly combined to instantiate a mapping-specific translator on the fly, enabling efficient and specialized translation without retraining. To capture long-range correspondence beyond local neighborhoods, we further reshape window tokens into a complementary grid-token layout and repeat the same cross-attention and FFN updates at a coarser granularity, enabling global information exchange with sparse structure. Finally, the updated tokens are merged back to a BEV feature map, producing the translated neighbor feature $\hat{F}_{j \rightarrow i}$ for downstream fusion.

**Training Procedure.** Stage 2 trains the Modality-Mapping Router (MMR) together with the Translator Parameter Bank (TPB) while keeping the modality-intrinsic encoder (MIE) frozen, as summarized in Alg. 2. At each iteration, we first sample a mini-batch of scenes $\{s_b\}_{b=1}^{B} \sim \mathcal{D}_{\text{tr}}$. For every agent $u$ in each scene, we sample a modality label $m_{b,u} \sim \pi(\mathcal{M}_{\text{tr}})$ and select the corresponding pretrained perception encoder $f_{m_{b,u}}^{\text{enc}} \in \mathcal{R}_{\text{tr}}$ to extract intermediate features $F_{b,u} = f_{m_{b,u}}^{\text{enc}}(X_{b,u})$. We then obtain the modality-intrinsic code $z_{b,u} = \text{MIE}(F_{b,u})$, where MIE is fixed and only provides a stable modality descriptor for subsequent mapping inference.

**Algorithm 2** Stage-2 Training of MMR and TPB

**Require:** Scene repository $\mathcal{D}_{\text{tr}}$; model repository $\mathcal{R}_{\text{tr}}$ covering $\mathcal{M}_{\text{tr}}$; pretrained (frozen) MIE; translator parameter bank $\text{TPB} = \{\Theta^{(k)}\}_{k=1}^K$; batch size $B$.

1: **while** not converged **do**
2:     Sample a mini-batch of scenes $\{s_b\}_{b=1}^B \sim \mathcal{D}_{\text{tr}}$
3:     Initialize pair set $\mathcal{P} \leftarrow \emptyset$
4:     **for** each scene $s_b$ **do**
5:        **for** each agent $u$ in $s_b$ **do**
6:           Sample modality $m_{b,u} \sim \pi(\mathcal{M}_{\text{tr}})$
7:           Assign encoder $f_{m_{b,u}}^{\text{enc}} \in \mathcal{R}_{\text{tr}}$
8:           $F_{b,u} \leftarrow f_{m_{b,u}}^{\text{enc}}(X_{b,u})$
9:           $z_{b,u} \leftarrow \text{MIE}(F_{b,u})$ {frozen}
10:        **end for**
11:        **for** each ego–neighbor pair $(i,j)$ in $s_b$ **do**
12:           Obtain aligned feature $\hat{F}_{j\rightarrow i}$ by geometric alignment
13:           $\delta_{j\rightarrow i} \leftarrow g(z_{b,j}, z_{b,i})$
14:           $\boldsymbol{\alpha}_{j\rightarrow i} \leftarrow \text{MMR}(\delta_{j\rightarrow i})$
15:           $\phi_{j\rightarrow i} \leftarrow \sum_{k=1}^K \alpha_{j\rightarrow i}^{(k)} \Theta^{(k)}$
16:           $F_{j\rightarrow i} \leftarrow \mathcal{T}_{\phi_{j\rightarrow i}}\left(\hat{F}_{j\rightarrow i}, F_{b,i}\right)$
17:           Add $(i, j, F_{j\rightarrow i})$ to $\mathcal{P}$
18:        **end for**
19:     **end for**
20:     Fuse and predict $\hat{Y}_i$ (using $\{F_{j\rightarrow i}\}_{(i,j,\cdot)\in\mathcal{P}}$), and compute the task loss $\mathcal{L}_{\text{task}}(\hat{Y}_i, Y_i)$
21:     Compute auxiliary losses and the total objective $\mathcal{L}_{\text{stage2}}$ according to Eq. (25)–(29)
22:     Update MMR and TPB by minimizing $\mathcal{L}_{\text{stage2}}$ in Eq. (29)
23: **end while**

Next, for each ego–neighbor pair $(i, j)$ within the sampled scenes, we first geometrically align the neighbor feature to the ego coordinate frame to obtain $\hat{F}_{j\rightarrow i}$. We compute a mapping descriptor $\delta_{j\rightarrow i} = g(z_{b,j}, z_{b,i})$ from the intrinsic codes of the source and target modalities, and feed it into MMR to predict the parameter-combination coefficients $\boldsymbol{\alpha}_{j\rightarrow i}$. Given $\boldsymbol{\alpha}_{j\rightarrow i}$, we instantiate a mapping-specific translator by linearly combining TPB experts. The instantiated translator $\mathcal{T}_{\phi_{j\rightarrow i}}$ takes the aligned neighbor feature and the ego feature as inputs, producing the translated feature

$$F_{j\rightarrow i} = \mathcal{T}_{\phi_{j\rightarrow i}}\left(\hat{F}_{j\rightarrow i}, F_{b,i}\right), \tag{31}$$

which is then collected for downstream fusion.

After all pairs in the mini-batch are processed, we fuse the translated features $\{F_{j\rightarrow i}\}$ with the ego pipeline to obtain predictions $\hat{Y}_i$, and compute the downstream task loss $\mathcal{L}_{\text{task}}(\hat{Y}_i, Y_i)$. We further compute the auxiliary losses defined in Eq. (25)–Eq. (29), including feature-level distillation, routing contrast, and router regularization, and

*Table 9.* Effect of intrinsic code dimension $d$.

| $d$ | AP@0.5 | AP@0.7 |
|---|---|---|
| 2 | 0.669 | 0.562 |
| 4 | 0.716 | 0.605 |
| 8 | 0.720 | 0.583 |
| 16 | 0.675 | 0.540 |
| 32 | 0.668 | 0.558 |
| 64 | 0.660 | 0.566 |

aggregate them into the total Stage 2 objective $\mathcal{L}_{\text{stage2}}$. Finally, we update only the parameters of MMR and TPB by minimizing $\mathcal{L}_{\text{stage2}}$, while all pretrained encoders in $\mathcal{R}_{\text{tr}}$ and the frozen MIE remain unchanged throughout Stage 2 training.

Upon completing Stage 2, we obtain a reusable TPB $\{\Theta^{(k)}\}_{k=1}^K$ together with a MMR that predicts mapping-specific combination coefficients. Together with the pretrained MIE from Stage 1, this forms an inference-time instantiation mechanism for open-world heterogeneous collaboration. Given an arbitrary source-to-target modality mapping $(m_j \rightarrow m_i)$ at test time, we first extract intrinsic codes $(z_j, z_i)$ via MIE and compute a mapping descriptor $\delta_{j\rightarrow i}$. MMR then outputs $\boldsymbol{\alpha}_{j\rightarrow i}$ to synthesize translator parameters by a lightweight linear combination, $\phi_{j\rightarrow i} = \sum_{k=1}^K \alpha_{j\rightarrow i}^{(k)} \Theta^{(k)}$, enabling on-the-fly construction of a mapping-specialized translator $\mathcal{T}_{\phi_{j\rightarrow i}}$. This design supports efficient any-to-any feature translation without retraining, while maintaining specialization across diverse and previously unseen modality pairs.

## C. More Experimental Analysis

### C.1. Effect of intrinsic code dimension.

We study the impact of the intrinsic code dimension $d$ in Eq. (10) on OPV2V-H to understand how the capacity of modality-intrinsic representations affects feature translation and downstream cooperative perception. As shown in Tab. 9, moderate dimensions achieve the best performance, with $d$=4 yielding the highest accuracy. When $d$ is too small (e.g., $d$=2), the intrinsic code lacks sufficient expressiveness to capture nuanced modality mappings, limiting translator instantiation quality and hurting detection accuracy. In contrast, overly large $d$ increases the intrinsic space volume and training difficulty: accurately structuring a high-dimensional intrinsic space requires substantially more modality samples and optimization budget to populate the space. With limited modality coverage, intrinsic codes can become nearly orthogonal and fail to form meaningful neighborhoods, which weakens the measurement of modality relations and degrades translation effectiveness, as

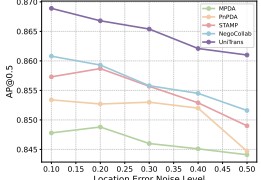 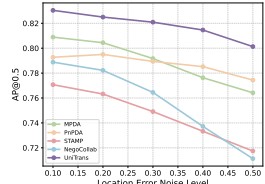 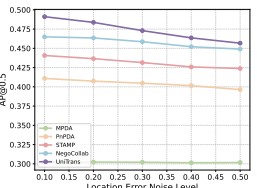

*Figure 3.* Perception performance under different location-error noise levels for four ego modalities.

reflected by the performance drop for a larger $d$. Overall, these results suggest that a compact yet expressive intrinsic space is crucial for stable mapping estimation and robust any-to-any translation.

## C.2. Modality organization of the intrinsic space.

To verify whether the learned intrinsic space is organized by modality rather than by scene-specific variations, we quantitatively compare the raw features and the intrinsic codes using silhouette scores (Rousseeuw, 1987; Kapse et al., 2024) and cosine-distance-based statistics. A higher silhouette score indicates stronger clustering under the given labels, while a larger cosine distance indicates stronger separation. As shown in Tab. 10, the modality silhouette score increases from 0.70 to 0.96 after Modality-Intrinsic Encoding, suggesting that the intrinsic codes form more compact modality-specific clusters. Meanwhile, the average distance between samples with the same modality but different scenes decreases substantially, indicating that scene-dependent variations are suppressed. In contrast, the distance between samples from the same scene but different modalities increases, showing that modality differences remain clearly separable even when the scene content is shared. These results demonstrate that the intrinsic embeddings are primarily organized by modality and remain stable across scene changes.

## C.3. Effect of TPB capacity.

Tab. 11 studies the impact of TPB capacity $K$, namely the number of parameter groups (in Eq. (22).) for translator instantiation. With small $K$ (e.g., $K=1$ or 2), the translator family is under-parameterized, limiting expressivity and resulting in weaker translation performance. Increasing $K$ improves performance and peaks at $K=8$, suggesting that a moderate capacity provides sufficient diversity for mapping-specific instantiation. Further increasing $K$ (e.g., 16 and 32) yields slightly lower accuracy, likely because larger capacity requires more training to adequately cover the expanded parameter space and to reliably learn routing, otherwise experts become less well trained and the instantiated translators generalize less effectively.

## C.4. Effect of the shared expert.

Tab. 12 studies the effect of the shared expert $\Theta^{(0)}$ in the TPB (Eq. (24)). Removing the shared expert leads to a clear performance drop, from $0.716/0.605$ to $0.666/0.573$, indicating that $\Theta^{(0)}$ is not merely a convenient design choice. Instead, it provides a common translation template that captures mapping-invariant knowledge, on top of which mapping-specific adjustments can be instantiated through the remaining expert bases. We further investigate whether using more shared experts is beneficial. Using two shared experts brings only marginal improvement at AP@0.5 but slightly lowers AP@0.7, while further increasing the number of shared experts degrades performance. This suggests that a single shared expert already provides a sufficient common basis, whereas averaging multiple shared bases may make optimization less stable under the same training budget.

## C.5. Robustness to localization errors.

Fig. 3 reports AP@0.5 under increasing localization-error noise for four ego modalities (from left to right: m3, m4, m7, and m27) on OPV2V-H. Across all modalities, the detection accuracy decreases smoothly as the noise level grows, indicating the expected sensitivity of cooperative perception to pose perturbations. Nevertheless, `UniTrans` exhibits a stable degradation trend and maintains reasonable performance over a wide noise range, suggesting that the learned translation and fusion pipeline is resilient to moderate localization errors commonly encountered in real-world deployments. This robustness is consistent across both LiDAR-dominant ego settings (m3/m4/m7) and the more challenging camera ego setting (m27), demonstrating that `UniTrans` can tolerate imperfect localization while preserving effective cross-agent collaboration.

## C.6. Extended Qualitative Verification

Fig. 4 visualizes intermediate features before and after translation, where the ego feature serves as the target-domain style that the neighbor feature should match. Compared with MPDA, `UniTrans` produces translated features that better preserve scene-relevant semantic structures while aligning more closely with the target-domain appearance, which is consistent with our design of dynamically instantiating

*Table 10.* Quantitative analysis of the modality organization of the intrinsic space. Distances are computed as cosine distances.

| Metric | Raw Feature | Intrinsic Code |
|---|---|---|
| Modality silhouette | 0.70 | **0.96** |
| Same modality, different scene distance | 9.42e-1 | **4.73e-4** |
| Different modality, same scene distance | 0.4187 | **1.0332** |

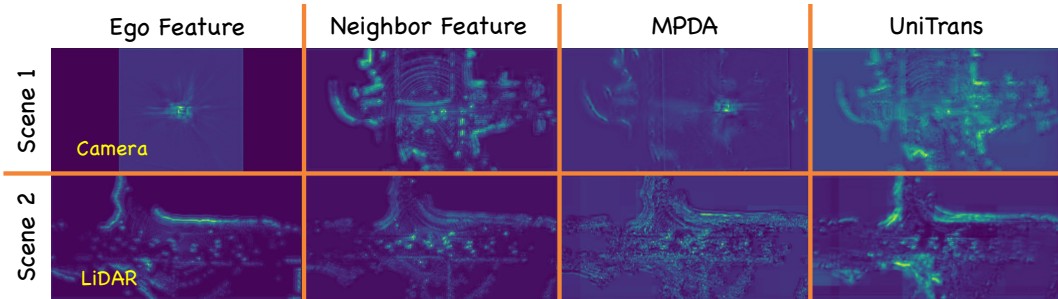

*Figure 4.* Qualitative visualization of any-to-any feature translation on OPV2V-H. *Ego Feature* denotes the target-domain representation, *Neighbor Feature* is the source feature to be translated, and the last two columns show the translated features produced by MPDA and `UniTrans` (camera ego in Scene 1 and LiDAR ego in Scene 2).

*Table 11.* Effect of TPB capacity $K$ in Eq. (22).

| $K$ | AP@0.5 | AP@0.7 |
|---|---|---|
| 1 | 0.608 | 0.502 |
| 2 | 0.652 | 0.556 |
| 4 | 0.708 | 0.598 |
| 8 (Ours) | 0.716 | 0.605 |
| 16 | 0.707 | 0.595 |
| 32 | 0.684 | 0.580 |

*Table 12.* Ablation study on the number of shared experts in the TPB on OPV2V. Each entry reports Avg. AP@0.5/AP@0.7.

| # Shared Experts | Avg. AP@0.5/AP@0.7 |
|---|---|
| 0 | 0.666 / 0.573 |
| 1 (Ours) | 0.716 / 0.605 |
| 2 | 0.718 / 0.601 |
| 4 | 0.701 / 0.591 |
| 8 | 0.675 / 0.568 |

a mapping-specific translator conditioned on the inferred neighbor-to-ego modality mapping. This advantage is particularly evident in cross-modality cases, where the modality gap is large and a generic mapping tends to lose fine-grained cues.

Fig. 5 further demonstrates that improved translation quality translates into better downstream cooperative perception. Across diverse scenes, `UniTrans` yields more accurate and stable 3D detections, with fewer missed objects and more consistent localization, indicating that mapping-conditioned translator instantiation provides more reliable target-consistent representations for collaborative detection.

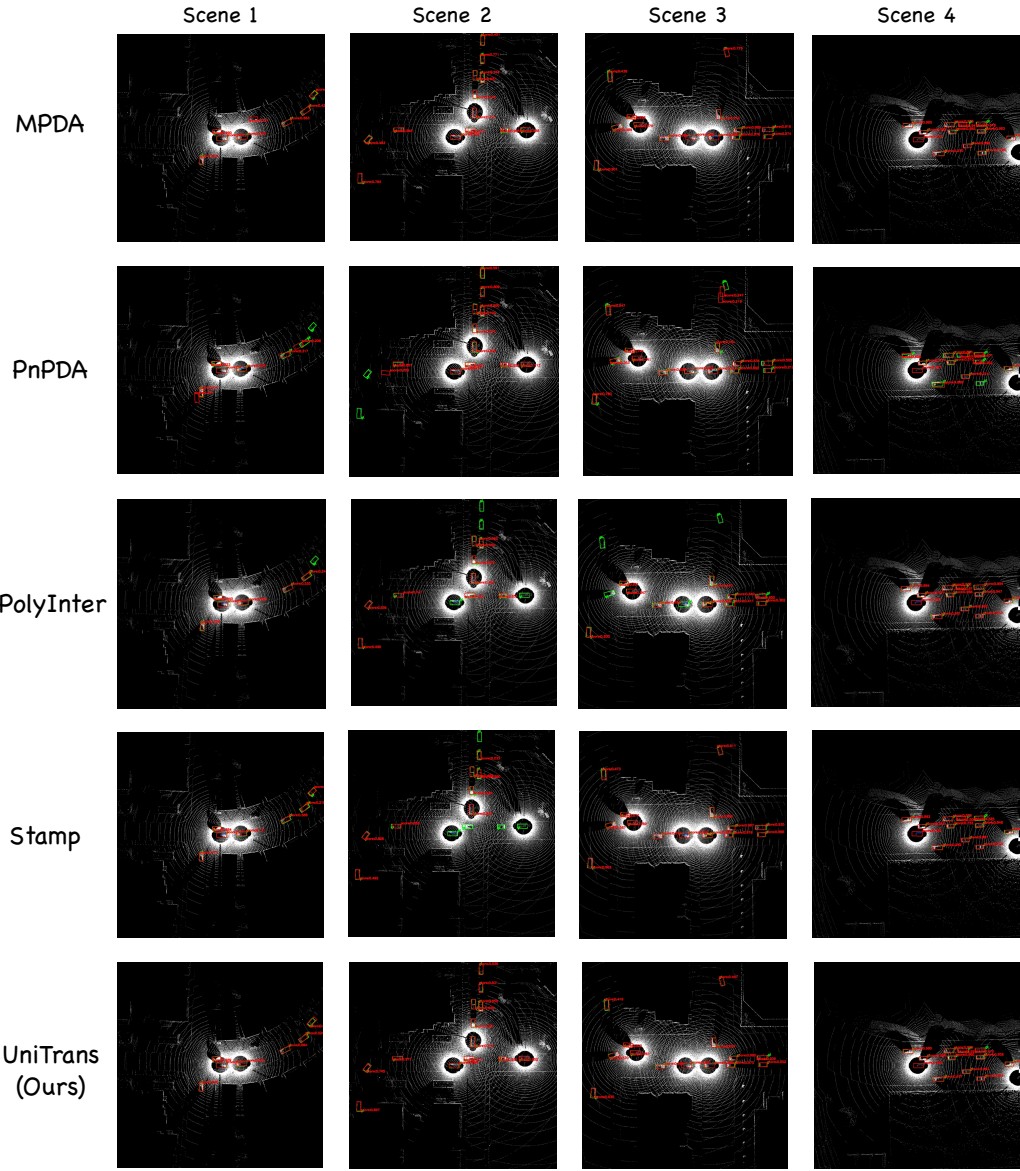

*Figure 5.* Qualitative cooperative detection results on OPV2V-H using translated features. We compare different translation methods across four scenes; predicted boxes are shown in red and ground-truth boxes in green.

