# OpenReview forum: "One Model to Translate Them All: Universal Any-to-Any Translation for Heterogeneous Collaborative Perception"
_ICML.cc/2026/Conference — ICML 2026 regular_

### Official Review · Reviewer_A8RU · 2026-03-06

**Soundness:** 2
**Presentation:** 3
**Significance:** 3
**Originality:** 3
**Overall Recommendation:** 4
**Confidence:** 4

**Summary:**

This paper proposes UniTrans, a universal translation model designed to address distribution discrepancies between intermediate features caused by heterogeneous sensors and encoder architectures—without requiring retraining. The key objective is to enable zero-shot any-to-any feature translation across modalities.
First, a Modality-Intrinsic Encoder is introduced to learn a latent space that captures inter-modality relationships. Based on this latent representation, the model conditionally composes parameters from a Translator Parameter Bank to instantiate a mapping-specific translator on the fly.
Through this mechanism, UniTrans enables effective cooperative perception even when new modalities are introduced, without requiring additional training.

**Compliance With Llm Reviewing Policy:**

Affirmed.

**Final Justification:**

The authors’ responses have clarified my concerns. Although the proposed method relies on sufficient diversity in pretrained backbones to ensure generality, the direction it suggests leads to meaningful performance improvements in the given scenario. Accordingly, I have decided to increase my score by one level.

**Key Questions For Authors:**

- It remains unclear whether the proposed method can maintain stable performance for emerging modalities that exhibit substantially larger distribution gaps from the training modalities. In the current experiments, the emerging modalities seem to be mostly within a limited range of structural variations. If the authors could provide additional discussion or empirical validation under more severe distribution shifts or fundamentally different encoder architectures, it could meaningfully affect the assessment of the paper in terms of soundness and significance.

- Table 3 provides an efficiency comparison against Classic MoE. However, adding computational cost comparisons with other translator-based methods beyond MPDA would make the evaluation more fair and comprehensive. Moreover, while the proposed method may be latency-efficient at inference time since it executes only a single instantiated translator, it may incur memory overhead because it needs to store K expert parameter sets. Could the authors provide a quantitative analysis of the total parameter count and memory footprint of the TPB (i.e., overall model size and storage requirements)?

- In standard MoE, each expert can often be interpreted as an independent model. In contrast, this paper learns and performs inference by linearly combining expert parameters. In this case, what semantic meaning do the learned experts have individually? For example, can they be interpreted as bases specialized for certain modality mappings, or should they be understood as more abstract translation primitives? It would be helpful to hear the authors’ perspective on how to interpret the experts.

- As part of the ablation study, could the authors compare MMR-based weighted composition against a simple uniform averaging baseline (e.g., combining all experts with equal weights)? In other words, evaluating a setting that removes MIE/MMR and constructs the translator solely via an averaged TPB composition would help isolate the practical contribution of routing.

- The proposed approach instantiates a single translator by forming a weighted sum of multiple expert parameters. This seems to require all expert models to share the same architecture and parameter shapes. Could the authors confirm whether this understanding is correct? If so, explicitly stating that such structural consistency is ensured (e.g., via geometric alignment and standardized feature formatting) would improve clarity for readers.

- In Tables 1 and 2, is the neighbor modality randomly sampled per example, or is it evaluated under fixed ego–neighbor modality pairs? If the modality pairing is fixed, it would be helpful to make this explicit in the table notation for clarity.

**Limitations:**

The paper does not provide sufficient analysis regarding the extent of distribution discrepancy between training and emerging modalities under which the proposed method can remain stable. In the current experimental setup, the emerging modalities appear to be largely confined to structural variations of the training modalities (e.g., changes in model capacity or voxel size).
Therefore, it remains unclear whether the framework can generalize equally well to modalities with fundamentally different encoder architectures or substantially different feature distributions compared to those observed during training. Additional theoretical discussion or empirical validation under more drastic distribution shifts would strengthen the claims of universality. More clearly delineating the boundary conditions of applicability would also help clarify the practical scope and real-world deployability of the proposed approach.

**Strengths And Weaknesses:**

# Soundness
**Strengths**
- The proposed method is organized into a two-stage framework—Stage 1 (MIE training) and Stage 2 (TPB–MMR-based translator learning)—with clearly defined objectives at each stage. The transition from learning modality-intrinsic representations in Stage 1 to instantiating mapping-conditioned translators in Stage 2 is logically coherent and well structured.
- The training design is systematic and carefully constructed, incorporating auxiliary losses such as feature distillation, routing contrast, and load balancing to jointly promote training stability and generalization.
- Experimentally, the paper evaluates the approach on both a simulation dataset (OPV2V-H) and a real-world dataset (DAIR-V2X), and includes comparisons with various baseline methods as well as ablation studies. Overall, the empirical evaluation is comprehensive and well executed.

**Weaknesses**
- The emerging modalities used in the experiments appear to differ from the training modalities primarily through structural variations (e.g., capacity changes, voxel size adjustments), rather than fundamentally distinct encoder architectures. As a result, the evaluation mainly validates generalization within a controlled variation range of existing models.
- It remains unclear whether the proposed framework would maintain similar generalization capability under more drastic distribution shifts, such as entirely different voxel resolutions or fundamentally different encoder designs. Consequently, the claimed “universal” applicability may not yet be fully substantiated across broader modality variations.

# Presentation
**Strengths**
- The paper clearly identifies the limitations of existing heterogeneous collaborative perception paradigms and naturally introduces the problem formulation of any-to-any translation as a principled extension. By contrasting with prior one-to-one adaptation and protocol-based approaches, the necessity of the proposed method is convincingly motivated.
- The overall pipeline—MIE → TPB → MMR—is described in a staged and structured manner, allowing readers to understand the role of each component and their interconnections. The narrative flow is systematic, and the methodological development is logically organized.

**Weaknesses**
- The inclusion of additional visualizations would further improve clarity and persuasiveness. For instance, illustrating the geometric properties of the intrinsic latent space—particularly how emerging modalities are embedded within it—or providing qualitative comparisons between translated features and target-domain features after alignment would help readers build more intuitive understanding.
- Other Issues
  - In Fig. 1(d), the conceptual distinction between “Scalability” and “Seamless Integration” is not clearly articulated, making the difference between the two terms somewhat ambiguous.
  - In Fig. 2(b), the relationship between S and E₁–Eₙ and the notation Θ(0), Θ(1)–Θ(k) is not explicitly clarified, which may cause interpretational confusion.
  - The font sizes in Table 1 and Table 2 are relatively large, resulting in slight visual imbalance; minor formatting adjustments could improve readability.
  - In Section 3.3, the embedding dimension d appears (line 268) without being explicitly defined in the preceding formulation, which may cause confusion for readers.

# Significance
**Strengths**
- The paper addresses an important scalability challenge that inevitably arises in real-world collaborative perception systems, where heterogeneous sensor configurations and encoder architectures coexist. Proposing a framework that can integrate new modalities without retraining is practically meaningful in realistic deployment scenarios.
- The zero-shot applicability—without additional fine-tuning or retraining—constitutes a significant contribution from the perspective of system scalability and maintenance cost.

**Weaknesses**
- As noted in the Soundness discussion, the emerging modalities used in the experiments appear to remain within the variation range of the original training modalities. Consequently, the true extent of the proposed method’s applicability under more drastic architectural or distributional shifts remains somewhat uncertain. Additional validation on fundamentally different encoder designs or modality distributions would strengthen the claims of universality.
- The analysis of computational overhead is relatively limited. For example, comparisons of computational cost with other translator-based methods beyond MPDA would improve clarity. Moreover, a quantitative breakdown of the total parameter count of the TPB and its memory requirements (overall model size) would provide a more complete assessment of practicality. A more comprehensive efficiency analysis—including FLOPs, memory footprint, and latency—would further reinforce the practical significance of the work.

# Originality
**Strengths**
- The work demonstrates meaningful originality by appropriately combining metric learning–based intrinsic representation learning with a MoE-style parameter combination mechanism and applying them to a new problem setting. In particular, unlike conventional MoE approaches that mix expert outputs, the proposed design linearly combines expert parameters to instantiate a mapping-specific translator, which differentiates the method by jointly considering efficiency and generalization.
- The construction of modality descriptors (e.g., channel statistics, Gram matrix) and the design of loss functions (e.g., contrastive loss, distillation loss, routing regularization) are convincingly presented as well-motivated adaptations of established techniques, restructured to fit the characteristics of the target problem.

**Weaknesses**
- The core components of the proposed framework—contrastive learning, MoE mechanisms, parameter interpolation, and feature distillation—are all based on widely studied existing techniques, and none individually introduces a fundamentally new theoretical framework.
- Therefore, the contribution is better characterized as a structured recombination and system-level integration of prior ideas tailored to a specific problem setting, rather than as a methodological breakthrough.

---

> ### Author Rebuttal · Authors · 2026-03-30
>
> We thank the reviewer for the thoughtful and detailed feedback, and we are encouraged that the reviewer recognizes the practical importance of the problem, the clear and coherent two-stage design, and the overall strength of the comprehensive experiments.
>
> **(1) Scope of  emerging modalities (q1, Soundness w1&w2, Significance w1).** (a) As shown in  Tab. 7, we already consider diverse encoder architectures (e.g., SECOND, VoxelNet), voxel resolutions (0.1, 0.4, and 0.8, which lead to substantially different feature resolutions), and backbones (e.g., ResNet and EfficientNet). These variations induce clear feature-distribution shifts, and our experiments show that UniTrans remains effective under such substantial heterogeneity and consistently outperforms prior methods. (b) Our goal is a universal model for any-to-any translator instantiation after sufficiently broad pretraining. As with large pretrained models, broader coverage leads to stronger zero-shot generalization. With such pretraining, the intrinsic latent space becomes more compact and well structured, enabling more accurate estimation of mappings for substantially different emerging modalities, more reliable expert-combination weights, and thus a better-matched translator. (c) We have added a study that reduces the number of training modalities. With fewer training modalities, the training set covers a narrower range of modality distributions, leading to larger train–test gaps at inference and thus lower performance, which further highlights the importance of sufficiently broad pretraining.
>
> |# training modalities|Avg. AP0.5/AP0.7|
> |-|-:|
> |4|0.564/0.486|
> |8|0.577/0.498|
> |16|0.668/0.576|
> |24 (Ours) |0.716/0.605|
>
> **(2) Expert interpretation (q3&q5).** The experts are best viewed as *shared translation bases* in the functional space of feature translators. A modality-to-modality translator is a mapping function, and the weighted combination of expert parameters instantiates a translator by composing these bases for a specific source-target pair. This reformulates complex cross-modality translation into learning shared bases plus their combination weights, which reduces problem difficulty and improves generalization. It requires all experts to share the same architecture and parameter shapes for parameters combination.
>
> **(3) Originality and Practical Novelty (Originality w1&w2).** UniTrans differs substantially from conventional techniques and is an original framework tailored to heterogeneous feature translation. It estimates modality mappings in the intrinsic latent space and reformulates complex translator functions as shared mapping bases wirth their linear combination, enabling zero-shot any-to-any translation for emerging modalities beyond prior methods.
>
> **(4) Efficiency(q2, Significance w2).** UniTrans is inference-efficient because it executes only one instantiated translator. The TPB is used only in Transformer linear layers and, with 1 shared expert + 4 experts, contains just 744,640 parameters (2.98 MB in FP32). The full model, including encoders, UniTrans, fusion, and task networks, uses about 3 GB GPU memory at inference. Other Transformer-based baselines also incur higher runtime overhead than MPDA, and we will add these results in the revision.
>
> **(5) Uniform averaging ablation (q4).** We added this control on OPV2V. Simple uniform parameter averaging performs close to the plain Transformer baseline, indicating that without routing the model largely degenerates to a conventional Transformer and underscoring the value of MMR-based dynamic routing.
>
> |Method|Avg. AP0.5/AP0.7|
> |-|-:|
> |Uniform averaging|0.588/0.496|
> |Transformer baseline|0.609/0.502|
> |UniTrans| 0.716/0.605|
>
> **(6) Evaluation setting (q6).** In Tables 1 and 2, ego–neighbor modality pairs are uniformly sampled on the test set to ensure a balanced number of scenes for each neighbor modality, and the sampled pairs are kept identical across all methods. This sampling protocol is consistent with that used in STAMP and NegoCollab. We will make this clearer in the paper.
>
> **(7) Presentation.** We will add more discussion of the intrinsic latent space geometry. Here, we provide quantitative evidence using silhouette and cosine-distance statistics: a higher silhouette indicates stronger clustering, and a larger cosine distance indicates stronger separation. After Modality-Intrinsic Encoding, samples with the same modality become nearly identical across scenes, while samples from the same scene but different modalities become more separated.
>
> |Metric|Raw feature|Intrinsic code|
> |-|-:|-:|
> |Modality silhouette |0.70|0.96|
> |Same modality, different scene distance |9.42e-1|4.73e-4|
> |Different modality, same scene distance |0.4187|1.0332|
>
> We will also clarify the distinction between *Scalability* (parameter/training complexity as modality types grow) and *Seamless Integration* (deployment-time onboarding of new agents without retraining), and revise the these issues noted by the reviewer.

---

> > ### Author Rebuttal · Reviewer_A8RU · 2026-04-01
> >
> > The authors’ responses have clarified my concerns. In particular, while the proposed method carries a condition that sufficient diversity in pretrained backbones is required to ensure its generality, I believe that the performance improvements demonstrated under this condition are still meaningful.

---

> > > ### Author Response · Authors · 2026-04-01
> > >
> > > We are truly pleased that our responses could clarify your concerns, and we are especially encouraged that you recognize the value of our work. This exchange has also helped us better understand and articulate both the technical details and the broader motivation of the paper, and it will directly help us further improve the manuscript. We sincerely appreciate your thoughtful feedback, and we hope that, with your help, this work can make a meaningful contribution to the community and support further progress in autonomous driving and collaborative perception. Thank you again.

---

### Official Review · Reviewer_N8vC · 2026-03-12

**Soundness:** 3
**Presentation:** 3
**Significance:** 2
**Originality:** 2
**Overall Recommendation:** 4
**Confidence:** 3

**Summary:**

This paper proposes a universal any-to-any feature modality translation model for heterogeneous collaborative perception. The goal is to integrate newly emerging agents with unseen intermediate feature modalities without retraining or fine-tuning. UniTrans uses a two-stage pretraining pipeline that learns a modality-intrinsic encoder to map intermediate features into a scene-invariant latent space. It also features a translator parameter bank with a router that predicts mapping-conditioned parameter-combination weights. The method is evaluated on several datasets with a constructed set of modality categories and held-out emerging modalities tested in a zero-shot setting.

**Compliance With Llm Reviewing Policy:**

Affirmed.

**Final Justification:**

The authors responded to my questions. With that said, I consider the paper overall in the borderline accept category, and will maintain my rating.

**Key Questions For Authors:**

1. How does UniTrans generalize when an emerging modality is substantially outside the training model repository, e.g., a backbone family not seen in pretraining rather than a depth/architecture variant of known encoders?

2. HEAL should be included as a baseline, or explicitly justify why it is not comparable under your evaluation protocol?

3. How are the key scalability parameters chosen, such as the number of experts K in the translator parameter bank and the intrinsic embedding dimension d? How sensitive are results to these choices?

4. The intrinsic encoder is intended to be scene-invariant instead of modality-specific. How do you verify that the intrinsic embeddings cluster primarily by modality rather than being driven by scene distribution shifts?

5. What happens if the ego modality itself is from the unseen set and both ego and neighbor are “emerging”? Is the intrinsic mapping and router still stable?

**Limitations:**

See weaknesses.

Minor issues:
- In the contribution list, the claim “improves perception performance by up to 10%” would be clearer to specify which metric, which dataset, and which baseline this improvement refers to.

**Strengths And Weaknesses:**

Strengths:
+ The paper targets a practical and well-motivated issue in collaborative perception

+ The method is technically coherent and reasonably well-structured. The modality-intrinsic latent space provides a mechanism for estimating modality mappings from limited test-time evidence, and the parameter-bank instantiation avoids multi-expert execution overhead at inference.

+ Experiments include both simulated and real-world datasets and compare against several relevant heterogeneous collaboration baselines and additional adapter/MoE variants.

+ The profiling results and ablations are useful for understanding efficiency and component contributions.



Weaknesses:

- The paper cites the open heterogeneous line of work, and in particular HEAL is highly relevant to the problem setting (even if HEAL still requires per-new-type training). It would strengthen the empirical story to include HEAL as a baseline or provide a clear reason why it is not comparable under the setup.

- The zero-shot open-world claim depends on strong pretraining assumptions. Stage 1 assumes access to a scene repository and a model repository that covers many modalities for pretraining the intrinsic encoder and routing behavior. It is not fully clear how realistic this is across manufacturers and how performance degrades when the training modality coverage is narrower.

- The feature distillation supervision in Stage 2 relies on constructing an ego-domain teacher by encoding neighbor observations with the ego encoder. While this is available during offline training, it is a fairly strong supervision signal and may partially explain gains. It would help to clarify how sensitive the method is when such supervision is weaker or less representative.

---

> ### Author Rebuttal · Authors · 2026-03-30
>
> We thank the reviewer for the careful reading and for recognizing the practical importance of the problem, the coherent two-stage design, and the usefulness of our profiling and ablation studies. We are encouraged that the reviewer views the paper as technically solid and relevant to real deployment.
>
> **(1) HEAL and comparability (w1&q2).** We agree that HEAL is highly relevant. However, real-world setting is always stricter: at deployment time, the emerging modalities are unseen during training, and no additional retraining or ground-truth support is available for adapting the encoder. Under this zero-shot open-world environment, HEAL has very limited generalization ability. We nevertheless added HEAL to the comparison:
>
> | | OPV2V | DAIR-V2X |
> |---|---:|---:|
> | HEAL | 0.493 / 0.395 | 0.423 / 0.310 |
> | Ours | **0.716 / 0.605** | **0.553 / 0.421** |
>
> **(2) Pretraining scope and narrower coverage (w2&q1).** We agree that strong zero-shot open-world performance requires sufficiently broad pretraining. In this sense, UniTrans follows the same scaling intuition commonly seen in foundation models: broader repositories lead to better coverage and stronger generalization. With sufficiently broad pretraining, the modality-intrinsic latent space becomes more compact and densely structured, enabling more accurate estimation of the mappings for emerging modalities as well as the mixture weights for translator instantiation.
>
> In practice, it is realistic that different manufacturers contribute partial data and model variants to build a shared pretraining repository, while still keeping their latest proprietary versions private.  Similar to commonly used LLMs such as LLaMA, ChatGPT, and Qwen, they achieve strong zero-shot generalization through large-scale pretraining with broad data coverage, following scaling laws.
>
> We also agree that narrower training coverage degrades performance. We have added a study by reducing the number of training modalities:
>
> | # training modalities | Avg. AP0.5/AP0.7 |
> |-|---:|
> |4 | 0.564 / 0.486 |
> |8 | 0.577 / 0.498 |
> |16 | 0.668 / 0.576 |
> |24 (Ours) | **0.716 / 0.605** |
>
> With fewer training modalities, the feature distribution gap at inference becomes larger; although performance degrades, UniTrans still outperforms prior methods that lack true zero-shot generalization to emerging modalities (see Ans.(2) of Reviewer G8Ay).
>
>
>
>
> **(3) Feature distillation supervision (w3).** We agree that $L_{\text{feat}}$ is a strong and helpful training-time supervision signal. For fairness, comparable baselines are also trained under the same supervision when applicable. We further performed an ablation by removing $L_{\text{feat}}$, and performance drops substantially:
>
> | $L_{\text{feat}}$ | Avg. AP0.5/AP0.7 |
> |---|---:|
> | w/  (Ours) | **0.716 / 0.605** |
> | w/o | 0.653 / 0.531 |
>
> This confirms that $L_{\text{feat}}$ acts as a stable teacher signal for aligning translated features to the ego domain, especially under large modality gaps, but it is not a privileged signal unique to our method.
>
>
> **(4) Why the intrinsic space is modality-organized rather than scene-driven (q4).** To verify this quantitatively, we compared raw features and intrinsic codes using silhouette[1,2] and cosine-distance-based statistics. A higher silhouette indicates stronger clustering, and a larger cosine distance indicates stronger separation. After Modality-Intrinsic Encoding, samples with the same modality become nearly identical even across different scenes, while samples from the same scene but different modalities become more separated:
>
> | Metric | Raw feature | Intrinsic code |
> |-|-:|-:|
> |Modality silhouette | 0.70 | **0.96** |
> |Same modality, different scene distance | 9.42e-1 | **4.73e-4** |
> |Different modality, same scene distance | 0.4187 | **1.0332** |
>
> These results support that the intrinsic embeddings are primarily organized by modality and remain stable across scene changes.
>
> **(5) Scalability hyperparameters K & d (q3).** These have already been studied in Appendix C.1 and C.2. In short, medium capacity works best: too small limits expressiveness, while too large makes learning more difficult. We will emphasize this more clearly in the main text.
>
> **(6) Both ego and neighbor emerging (q5).** Yes. Tables 1 and 2 already evaluate this setting, and the strong results indicate that the intrinsic mapping and router remain stable. We will make this point more explicit in the revision.
>
> **(7) Minor clarification.** The statement “improves perception performance by up to 10%” refers to the gain over STAMP on OPV2V. We will clarify this explicitly in the revision.
>
>
> [1] Rousseeuw, Silhouettes: A Graphical Aid to the Interpretation and Validation of Cluster Analysis, J. Comput. Appl. Math, 1987.
>
> [2] Kapse et al., SI-MIL: Taming Deep MIL for Self-Interpretability in Gigapixel Histopathology, CVPR, 2024.

---

> > ### Author Rebuttal · Reviewer_N8vC · 2026-04-01
> >
> > The rebuttal addresses most of my concerns with added results. The remaining concern is that zero-shot performance may be limited by how broad and representative the pretraining repository is, though the rebuttal scopes this. I will maintain my rating.

---

> > > ### Author Response · Authors · 2026-04-01
> > >
> > > We are very pleased that our additional explanations and results addressed most of your concerns.
> > > Regarding the remaining concern, we would like to clarify why UniTrans remains effective and practically meaningful, and why such broad and representative pretraining is achievable in practice.
> > >
> > > (1) Our current setup **already covers substantial heterogeneity** rather than only mild variations. As shown in Tab. 7, we consider diverse encoder architectures (e.g., SECOND, PointPillar, VoxelNet, and LSS), voxel resolutions (0.1, 0.4, and 0.8, which lead to substantially different feature resolutions), and backbones (e.g., ResNet and EfficientNet, with features extracted from different stages). These choices induce *clear feature-distribution shifts*, and UniTrans still remains effective and consistently outperforms prior methods under such heterogeneity.
> > >
> > > (2) Our goal is a universal model for any-to-any translator instantiation *after sufficiently broad pretraining*. Similar to the scaling behavior [1,2] observed in large pretrained models, broader coverage leads to stronger zero-shot generalization. In UniTrans, broader pretraining makes the intrinsic latent space more compact and better structured, with denser modality anchors and a more stable latent manifold. This allows an emerging modality to be localized more accurately in the intrinsic space, which in turn improves estimation of source-to-target modality mappings. At the same time, the router sees a richer set of “mapping $\rightarrow$ expert-combination” patterns during training, which improves zero-shot routing.
> > >
> > > Fundamentally, UniTrans **simplifies heterogeneous feature translation by reformulating a complex translator function as a linear combination of shared mapping bases, making the learning problem easier to optimize and more generalizable**.
> > >
> > > (3) We also believe this requirement is **practically achievable**. With the rapid development of autonomous driving and collaborative perception, diverse sensor setups, encoder architectures, and large-scale datasets are becoming increasingly available[4,5,6]. As a result, it is realistic to build, over time, a sufficiently broad and representative pretraining repository, making UniTrans a deployable solution rather than a purely conceptual one.
> > >
> > > (4) Our additional study directly supports this trend. Reducing the number of training modalities indeed enlarges the train–test gap and degrades performance, but **UniTrans still remains substantially stronger than prior heterogeneous translation methods** in the real-world zero-shot setting:
> > >
> > > | Training modalities | Avg. AP0.5/0.7 |
> > > |-| - |
> > > |4| 0.564 / 0.486|
> > > |8| 0.577 / 0.498|
> > > |16| 0.668 / 0.576|
> > > |24 (Ours) |0.716 / 0.605|
> > >
> > > Our experimental results are also consistent with the scaling-law trend [1,3].
> > > For reference, prior methods under the same zero-shot setting perform much worse:
> > >
> > > |Method| Avg. AP0.5/0.7 |
> > > |-|-|
> > > |MPDA| 0.393 / 0.322 |
> > > |PnPDA| 0.388 / 0.310 |
> > > |PolyInter  | 0.391 / 0.330|
> > > |STAMP| 0.451 / 0.384|
> > > |NegoCollab | 0.495 / 0.413 |
> > > |Ours| 0.716 / 0.605|
> > >
> > > This shows that while broader and more representative pretraining is important, **UniTrans is already a major step beyond prior heterogeneous feature translation methods for real-world zero-shot deployment**.
> > >
> > > (5) We agree that determining **how broad and representative the pretraining repository should be is ultimately a challenging systems and engineering question**, much like for LLMs and VLMs such as ChatGPT, Qwen, and LLaMA, whose strong zero-shot performance also depends on large-scale pretraining [7,8]. This remains an open challenge even for today’s leading foundation-model developers[9]. We therefore view it **not as a weakness unique to UniTrans**, but as a natural frontier for open-world collaborative perception.
> > >
> > >
> > > We sincerely appreciate your thoughtful assessment and are glad that the rebuttal helped clarify the scope and value of our work.
> > >
> > >
> > > [1] Radford et al., Learning Transferable Visual Models From Natural Language Supervision, ICML 2021.
> > >
> > > [2] Kaplan et al., Scaling Laws for Neural Language Models, CoRR, 2020.
> > >
> > > [3] Gadre et al., Language Models Scale Reliably with Over-Training and on Downstream Tasks, ICLR, 2025.
> > >
> > > [4] Yazgan et al., A Survey on Intermediate Fusion Methods for Collaborative Perception Categorized by Real World Challenges, arXiv, 2024.
> > >
> > > [5] Wan et al., Systematic Literature Review on Vehicular Collaborative Perception—A Computer Vision Perspective, arXiv, 2025.
> > >
> > > [6] Bai et al., A Survey and Framework of Cooperative Perception, IEEE Transactions on Intelligent Transportation Systems, 2024.
> > >
> > > [7] Cherti et al., Reproducible Scaling Laws for Contrastive Language-Image Learning, CVPR 2023.
> > >
> > > [8] Chen et al., PaLI: A Jointly-Scaled Multilingual Language-Image Model, ICLR 2023.
> > >
> > > [9] Li et al., DataComp-LM: In Search of the Next Generation of Training Sets for Language Models, NeurIPS 2024

---

### Official Review · Reviewer_G8Ay · 2026-03-13

**Soundness:** 2
**Presentation:** 3
**Significance:** 2
**Originality:** 2
**Overall Recommendation:** 4
**Confidence:** 4

**Summary:**

This paper studies open-world zero-shot feature translation for heterogeneous collaborative perception. The core problem is the following: when a newly connected vehicle or roadside unit produces intermediate features in a modality that is inconsistent with the existing system, and joint retraining or finetuning is no longer feasible, how can the system still translate the neighbor feature into a representation space that is compatible with the ego agent’s fusion module? To address this, the paper proposes UniTrans. The key idea is to first estimate the mapping from source modality to target modality in a modality-intrinsic latent space, and then use this mapping to instantiate a dedicated translator on demand.

**Compliance With Llm Reviewing Policy:**

Affirmed.

**Key Questions For Authors:**

See weakness

**Limitations:**

See weakness

**Strengths And Weaknesses:**

### Strengths

1. **The paper targets a practically important deployment problem.**

    A key strength is that it focuses on a genuinely difficult issue in heterogeneous collaborative perception: when a new modality enters the system, retraining the whole model is often unrealistic. This is more practically relevant than improving AP within a fixed and closed modality set.

2. **The method decomposition is clear and internally coherent.**

    The framework is well structured: MIE is responsible for modality localization, MMR and TPB are used to instantiate a translator, and MCT performs the actual feature translation. The pipeline is easy to follow, and each stage is supported by a corresponding training objective. Overall, the design forms a relatively complete closed loop.


### Weaknesses

1. **Some important ablations are missing.**

    The paper treats the shared expert Theta as an important design component, but does not provide an ablation showing how performance changes without it. This makes it difficult to assess whether the shared expert is truly necessary or just a convenient design choice.

2. **The fairness of the baseline comparison is somewhat unclear.**

    To satisfy the zero-shot any-to-any setting, the authors modify or adapt methods such as STAMP, NegoCollab, and ConvNeXt through structural changes or unified pretraining. This is understandable given the problem formulation, but it also means that the compared baselines are not evaluated in their original best-performing setups. As a result, it is hard to determine whether their weaker performance is due to intrinsic method limitations or because they are being evaluated outside their native design assumptions.

---

> ### Author Rebuttal · Authors · 2026-03-30
>
> We thank the reviewer for recognizing the practical importance of the problem and the clear, closed-loop design of UniTrans. We are encouraged that the reviewer views the paper as technically solid and relevant to real deployment.
>
> **(1) Shared expert (w1).** We agree that its necessity should be verified explicitly. We therefore added an ablation on OPV2V, shown below. The results indicate that having a shared expert is indeed beneficial, as it provides a common translation template on top of which mapping-specific adjustments can be instantiated.
>
> Further, we investigate whether using a larger number of shared experts is beneficial. Increasing the number of shared experts brings little gain and can even hurt optimization (under the same training epochs), likely because averaging multiple shared bases makes training less stable.
>
> | # shared experts | Avg. AP0.5/AP0.7 |
> |---|---|
> | 0 | 0.666 / 0.573 |
> | 1 (Ours) | **0.716 / 0.605** |
> | 2 | 0.718 / 0.601 |
> | 4 | 0.701 / 0.591 |
> | 8 | 0.675 / 0.568 |
>
> **(2) Fairness of baseline comparison (w2).** We appreciate this concern. As shown in Appendix A.4, we already *strengthened* prior methods so that they can better fit the strict zero-shot open-world inference setting, because their original forms perform much worse. To make this even clearer, we additionally report results when these methods are trained under their *native assumptions*, training feature translators for specific modality settings rather than zero-shot any-to-any deployment. Under these setup, they generalize poorly to newly emerging modalities:
>
> | Method | OPV2V | DAIR-V2X |
> |---|---|---|
> | MPDA | 0.393 / 0.322 | 0.367 / 0.289 |
> | PnPDA | 0.388 / 0.310 | 0.355 / 0.291 |
> | PolyInter | 0.391 / 0.330 | 0.370 / 0.278 |
> | STAMP | 0.401 / 0.334 | 0.389 / 0.307 |
> | NegoCollab | 0.431 / 0.351| 0.401 / 0.322 |
> | **UniTrans** | **0.716 / 0.605** | **0.553 / 0.421** |
>
> These results suggest that the gap is not merely due to evaluating baselines outside their preferred setup. Rather, prior translator-based and negotiation-based methods are not architecturally designed for *deployment-time zero-shot adaptation to emerging modalities*. In contrast, UniTrans is built precisely for this requirement: it estimates the source-target modality mapping in the intrinsic space and then dynamically instantiates a mapping-specific translator, enabling scalable and seamless integration of new agents without retraining.

---

> > ### Author Rebuttal · Reviewer_G8Ay · 2026-04-03
> >
> > I greatly appreciate the authors' detailed rebuttal and clarifications. I have decided to keep my original score unchanged.

---

> > > ### Author Response · Authors · 2026-04-04
> > >
> > > We sincerely appreciate the thoughtful reviews and helpful suggestions that have strengthened our work.

---

### Decision · Program_Chairs · 2026-04-30

**Decision:**

Accept (regular)

**Comment:**

The submission addresses a valid problem in collaborative perception around what to do when a new modality enters the system. Simplistic approaches, such as retraining the entire model, are often too expensive or ineffective. The proposed method forms a mapping from source to target modality in a modality-intrinsic latent space and uses that to instantiate a dedicated translator.

All reviewers are in support of the submission. The reviewers were specifically asked to comment 1) if the paper is technically sound and 2) why they'd find it exciting. The rebuttal was also incorporated. No fundamental flaw was uncovered in the review process. While the reviewers stopped short of finding the submission transformational, with a level of novelty and applicability that would likely lead to a change in common practices, it has enough merit to warrant exposure at ICML. @the authors, please carefully take into account all reviewers' comments and exchanges, and address them in the camera-ready for a greater impact of the paper.